



# Simulation of fine organic aerosols in the western Mediterranean area during the ChArMEx 2013 summer campaign

Arineh Cholakian[1,2], Matthias Beekmann[1], Augustin Colette[2], Isabelle Coll[1], Guillaume Siour[1], Jean Sciare[3,5], Nicolas Marchand[4], Florian Couvidat[2], Jorge Pey[4*], Valerie Gros[3], Stéphane Sauvage[6**], Vincent Michoud[9], Karine Sellegri[7], Aurélie Colomb[7], Karine Sartelet[8], Helen Langley DeWitt[4], Miriam Elser[10***], André S. H. Prévot[10], Sonke Szidat[11], François Dulac[3]

1 Laboratoire Inter-Universitaire des Systèmes Atmosphériques (LISA), UMR CNRS 7583, Université Paris Est Créteil et Université Paris Diderot, Institut Pierre Simon Laplace, Créteil, France (arineh.cholakian@lisa.u-pec.fr)
2 Institut National de l'Environnement Industriel et des Risques, Parc Technologique ALATA, Verneuil en Halatte, France
3 Laboratoire des Sciences du Climat et de l'Environnement, LSCE/IPSL, CEA-CNRS-UVSQ, Université Paris-Saclay, Gif-sur-Yvette, France
4 Aix-Marseille Université, CNRS, LCE FRE 3416, Marseille, 13331, France
5 The Cyprus Institute, Energy, Environment and Water Research Center, Nicosia, Cyprus
6 IMT Lille Douai, Univ. Lille, Département Sciences de l'Atmosphère et Génie de l'Environnement, F-59000 Lille, France
7 LAMP, Campus universitaire des Cezeaux, 4 Avenue Blaise Pascal, 63178 Aubière, France
8 CEREA, Joint Laboratory École des Ponts ParisTech – EDF R & D, Université Paris-Est, 77455 Marne la Vallée, France
9 Mines Douai, Département Sciences de l'Atmosphère et Génie de l'Environnement (SAGE), 59508 Douai, France
10 Paul Scherrer Institute, 5232 Villigen–PSI, Switzerland
11 University of Bern, Freiestrasse 3, CH-3012 Bern, Switzerland
* Now at the Spanish Geological Survey, IGME, 50006 Zaragoza, Spain
** Now at Mines Douai, Département Sciences de l'Atmosphère et Génie de l'Environnement (SAGE), 59508 Douai, France
*** Laboratory for Advanced Analytical Technologies, Empa, Dübendorf, 8600, Switzerland

**Abstract.** The simulation of fine organic aerosols with CTMs (Chemistry Transport Models) in the western Mediterranean basin has not been studied until recently. The ChArMEx (the Chemistry-Aerosol Mediterranean Experiment) SOP 2 (Special Observation Period 2) intensive field campaign in summer of 2013 gathered a large and comprehensive dataset of observations allowing the study of different aspects of the Mediterranean atmosphere including the formation of organic aerosols (OA) in 3D models. In this study, we used the CHIMERE CTM to perform simulations for the duration of the SAFMED (Secondary Aerosol Formation in the MEDiterranean) period (July to August 2013) of this campaign. In particular, we evaluated four schemes for the simulation of OA, including the CHIMERE standard scheme, the VBS (Volatility Basis Set) standard scheme with two parameterizations including aging of biogenic secondary OA, and a modified version of the VBS scheme which includes fragmentation and formation of non-volatile OA. The results for these four schemes are compared to observations at two stations in the western Mediterranean basin, located in Cap Corse (Corsica) and Cap Es Pinar (Mallorca). These observations include OA mass concentration, PMF (positive matrix factorization) results of different OA fractions, and $^{14}$C observations showing the fossil or non-fossil origins of carbonaceous particles. It is concluded that the modified VBS scheme is close to observations in all three aspects mentioned above; the standard VBS scheme without BSOA (Biogenic Secondary Organic Aerosol) aging also has a satisfactory performance in simulating the mass concentration of OA, but not for the source origin analysis comparisons. In addition, the OA sources over the western Mediterranean basin are explored. OA shows a major biogenic origin,





especially at several hundred meters height from the surface; however over the Gulf of Genoa near the surface, the anthropogenic origin is of similar importance. A general assessment of other species was performed to evaluate the robustness of the simulations for this particular domain before evaluating OA simulation schemes. It is also shown that the Cap Corse site presents important orographic complexity which makes comparison between model

simulations and observations difficult. A method was designed to estimate an orographic representativeness error for a list of species and yields an uncertainty of between 50-85% for primary pollutants, and around 2-10% for secondary species, for these species model to observations comparisons are only little impacted by orography.

## 1 Introduction

The Mediterranean basin is subject to multiple emission sources; anthropogenic emissions that are transported

from adjacent continents or are produced within the basin, local or continental biogenic and natural emissions among which the dust emissions from northern Africa can be considered as an important source (Pey et al., 2013 ; Vincent et al., 2016). All these sources, the geographic particularities of the region favoring accumulation of pollutants, the prevailing meteorological conditions favorable for intense photochemistry and thus secondary aerosol formation, make the Mediterranean area a region experiencing a heavy burden of aerosols (Monks et al.,

2009 ; Nabat et al., 2012). In the densely populated coastal areas, this aerosol burden constitutes a serious sanitary problem considering the harmful effects of fine aerosols on human health (Martinelli et al., 2013).  In addition, studies have shown that the Mediterranean area could be highly sensitive to future climate change effects (Giorgi, 2006 ; Lionello and Giorgi, 2007). This could affect aerosol formation processes, but in turn the aerosol load also affects regional climate (Nabat et al., 2013). These interactions, the high aerosol burden in the area, and its health

impact related to the high population density situated around the basin makes this region particularly important to study.

The aforementioned primary emissions can be in form of gaseous species, as particulate matter, or as semi-volatile species distributed between both phases (Robinson et al., 2007a). In the atmosphere, they can subsequently undergo complex chemical processes lowering their volatility to form secondary particles. The processes are not

entirely elucidated especially for the formation of SOA (Secondary Organic Aerosols, for example (Kroll and Seinfeld, 2008), from initially emitted biogenic or anthropogenic VOC (Volatile Organic Compounds) and SVOC (semi-volatile organic compounds).

The chemical composition of aerosols has been studied in detail in the eastern Mediterranean area (Lelieveld et al., 2002 ; Bardouki et al., 2003 ; Sciare et al., 2005 ; Koçak et al., 2007 ; Koulouri et al., 2008 ; Sciare et al., 2008)

and to a less extent in the western part (Sellegri et al., 2001 ; Querol et al., 2009 ; Minguillón et al., 2011 ; Ripoll et al., 2014 ; Menut et al., 2015 ; Arndt et al., 2017). Little focus has been given on the formation of organic aerosol (OA) over the western Mediterranean even if OA can play a significant role in both local and global climate (Kanakidou et al., 2005)  and can affect health (Pöschl, 2005 ; Mauderly and Chow, 2008). Its contribution has been calculated in studies to be nearly 30% in $PM_1$ for the eastern Mediterranean area during the FAME 2008

campaign (i.e. Hildebrandt et al., 2010). It is also important to know the contribution of different sources (biogenic, anthropogenic) to the total concentration of OA in the western part of the basin. Such studies have been performed for the eastern Mediterranean basin (Hildebrandt et al., 2010), while for the western part of the basin, they have been in general restricted to coastal cities such as Marseilles and Barcelona (El Haddad et al., 2011 ; Mohr et al., 2012 ; Ripoll et al., 2014).





The ChArMEx (the Chemistry Aerosol Mediterranean Experiment; http://charmex.lsce.ipsl.fr) project was organized in this context, with a focus over the western Mediterranean basin, in order to better assess the sources, formation, transformation and mechanisms of transportation of gases and aerosols. During this project, detailed measurements were acquired not only for the chemical composition of aerosols, but also for a large number of

gaseous species from both ground-based and airborne platforms. The project ChArMEx is divided into different sub-projects, each with a different goal; among those, the SAFMED (Secondary Aerosol Formation in the MEDiterranean) project aimed at understanding and characterizing the concentrations and properties of OA in the western Mediterranean (for example Nicolas, 2013 ; Di Biagio et al., 2015 ; Chrit et al., 2017 ; Arndt et al., 2017 ; Freney et al., submitted, 2017 ; Pey et al., 2017 in prep). To reach these goals, two intense ground-based and

airborne campaigns were organized during July-August of 2013 and also summer of 2014. The focus of the present study is on the SAFMED campaign in summer of 2013, since detailed measurements on the formation of OA and precursors were obtained during this period namely at Ersa (Corsica) and Es Pinar (Mallorca). Other ChArMEx sub-projects and campaigns included the TRAQA (Transport et Qualité de l'Air) campaign in summer 2012 set-up to study the transport and impact of continental air on atmospheric pollution over the basin (Sič et al., 2016),

the ADRIMED (Aerosol Direct Radiative Impact in the Mediterranean, June –July 2013) campaign aiming at understanding and assessing the radiative impact of various aerosol sources (Mallet et al., 2016).

Modelling of aerosol processes and properties is a difficult task. Aside from the lack of knowledge of aerosol formation processes, the difficulty lies in the fact that organic aerosols present an amalgam of thousands of different species that cannot all be represented in a 3D chemistry transport model due to limits in computational

resources. Therefore a small number of lumped species with characteristics that are thought to be representative of all the species in each group are used instead. There are many different approaches that can be used in creating representative groups for organic aerosols (e.g. which characteristics to use to group the species, which species to lump together, physical processes that should be presented for their simulation…). It is therefore necessary to test these different simulation schemes for organic aerosols in different regions and compare the results to experimental

data to check for their robustness. In this work, different configurations of the Volatility Basis Set (Donahue et al., 2006 ; Robinson et al., 2007 ; Shrivastava et al., 2013) and the base parameterization of the CHIMERE 3D model (Bessagnet et al., 2008 ; Menut et al., 2013) are used for this purpose. Although Chrit et al. (2017) modelled SOA formation in the western Mediterranean during the ChArMEx summer campaigns, they used a surrogate approach rather than a VBS approach in their air-quality modeling platform. The simulated concentrations and properties

(oxidation and affinity with water) of organic aerosols agree well to the observations performed at Ersa (Corsica), after they had modified their model to include the formation of extremely low volatility organic aerosols and organic nitrate from monoterpenes oxidation.

The present study focuses on the comparison of different OA formation schemes implemented in the CHIMERE chemistry-transport model for simulation of OA over the western Mediterranean area, with an extensive

experimental data pool obtained during the SAFMED campaign. The comparisons aim at assessing the robustness of each scheme with regard to different criteria as mass, fossil and modern fraction, volatility. Section 2 describes the model and the inputs used for the simulations. Also, the evaluated schemes are explained in this section in more detail. The experimental data used for simulation-observation comparisons are discussed in section 3. An overall validation of the model is presented in section 4, together with comparisons of different gaseous and

particulate species and meteorological parameters to observations. In section 5, comparison of implemented



schemes to measurements regarding concentration, oxidation state and origin of organic aerosol are presented. In section 6, the contribution of different sources to the organic aerosol present in the basin is explored, before the conclusion in section 7.

## 2 Model setup

The CHIMERE model (Menut et al., 2013; http://www.lmd.polytechnique.fr/chimere) is an offline regional CTM (Chemistry-Transport Model) which has been tested rigorously for Europe and France (Zhang et al., 2013 ; Petetin et al., 2014; Colette et al., 2015; Menut et al., 2015 ; Rea et al., 2015). It is also widely used in both research and forecasting activities in France, Europe and other countries (Hodzic and Jimenez, 2011). In this work, a slightly modified version of the CHIMERE 2014b configuration is used to perform the simulations. The modifications

concern an updated description of the changes in aerosol size distribution due to condensation and evaporation processes (Mailler et al., 2016). Four domains are used in the simulations; a coarse domain covering whole Europe and Northern Africa with a 30km resolution, and three nested domains inside the coarse domain with resolutions of 10km, 3km and 1km (figure 1, table 1). The 10km resolved domain covers the western Mediterranean area and the two smaller domains are centered on the Cap Corse, where the main field observations in SAFMED were

performed.  Such highly resolved domains are necessary to resolve the complex orography of Cap Corse ground-based measurements site which will be discussed in section 4, while for the flatter Es Pinar site, a 10km resolution is sufficient. The simulations for each domain contain 15 vertical levels starting from 50m to about 12km asl (vertical resolution of an average of 400m within the continental boundary layer (CBL) and 1km above CBL). The CHIMERE model needs as mandatory inputs a set of gridded data: meteorological data, emission data for both

biogenic and anthropogenic sources, landuse parameters, boundary/limit conditions and other optional inputs such as dust and fire emissions. Given these inputs, the model produces the concentrations and deposition fluxes for major gaseous and particulate species and also intermediate compounds.

Meteorological inputs are calculated with WRF (Weather Research Forecast) regional model (Wang et al., 2015) forced by NCEP (National Centers of Environmental Predictions) reanalysis meteorological data

(http://www.ncep.noaa.gov) with a base resolution of 1°. Slightly larger versions of each domain with the same horizontal resolutions (Table 1) are used for the meteorological simulations. The WRF configuration used for this study  consists of the Single Moment-5 class microphysics scheme (Hong et al., 2004), the RRTMG radiation scheme (Mlawer et al, 1997), the Monin-Obukhov surface layer scheme (Janjic, 2003), and the NOAA Land Surface Model scheme for land surface physics (Chen et al., 2001). Sea surface temperature update, surface grid

nudging (Liu et al., 2012, Bowden et al., 2012), ocean physics and topographic wind options are activated. Also, the feedback option is activated, meaning that the simulation of the nested domains can influence the parent domain. Some observation-simulation comparisons are presented for meteorological parameters in section 4.

Anthropogenic emissions for all but shipping snap sectors come from the HTAP-V2 (Hemispheric Transport of Air Pollution, Surendran et al., 2015, http://edgar.jrc.ec.europa.eu/htap_v2/index.php) inventory. The shipping

sector in this inventory was judged to overestimate ship traffic around the Cap Corse area, especially on the shipping lines between Marseilles and Corsica Island, due to overweighing ferries with respect to cargos, which can be explained by the fact that boat traffic description is based on voluntary information. Therefore HTAP-V2 shipping emissions were replaced by those of the MACC-III inventory (Kuenen et al., 2014). The base resolution



of HTAP inventory is 10km×10km and that of MACC-III inventory is 7km×7km. For both inventories the emissions for the year 2010 were used, since this year was the latest common year in the two inventories.

Biogenic emissions are calculated using MEGAN (Model of Emissions of Gases and Aerosols from Nature, Guenther et al., 2006) including isoprene, limonene, α-pinene, β-pinene, ocimene and other monoterpenes with a base horizontal resolution of 0.008°×0.008°. The land-use data comes from Globcover (Arino et al., 2008) with a base resolution of 300m×300m. Initial and boundary conditions of chemical species are taken from the climatological simulations of LMDz-INCA3 (Hauglustaine et al., 2014) for gaseous species and GOCART (Chin et al., 2002) for particulate matter.

The chemical mechanism used for the baseline gas-phase chemistry is the MELCHIOR2 scheme (Derognat et al., 2003). This mechanism has around 120 reactions to describe the whole gas-phase chemistry.

The CHIMERE aerosol module is responsible for the simulation of physical and chemical processes that influence the size distribution and chemical speciation of aerosols (Bessagnet et al., 2008). This module distributes aerosols in a number of size bins, here 10 bins ranging from 40nm to 40μm. The module also addresses coagulation, nucleation, condensation, as well as dry and wet deposition processes. The basic chemical speciation includes EC, sulfate, nitrate, ammonium, SOA, dust, salt and PPM (primary particulate matter other than ones mentioned above).

### 2.2 Organic aerosol simulation

The SOA particles are divided, depending on their precursors, into two groups: ASOA (anthropogenic SOA), and BSOA (biogenic SOA). Four schemes were tested to simulate their formation, more detail on each scheme is presented below.

### 2.2.1 CHIMERE standard scheme

The scheme for simulation of SOA in CHIMERE (Bessagnet et al., 2008) consists of a single-step oxidation process, where VOC lumped species are directly transformed into SVOC (Semi-volatile Organic Compounds) with yields that are taken from experimental data (Pun and Seigneur, 2007) and then are distributed into gaseous and particulate phases (figure 2-a) following the partitioning theory of Pankow (Pankow, 1987). The precursors for this scheme are presented in the supplementary information. A number of 11 semi-volatile surrogate compounds are formed from these precursors, which include six hydrophilic species, three hydrophobic species and two surrogates for isoprene oxidation. The sum of all 11 species results in the concentration of simulated SOA in this scheme.

### 2.2.2 VBS scheme

As an alternative to single step schemes like the one in CHIMERE, the VBS (volatility basis set) approach was developed. In these types of schemes, SVOC are divided into volatility bins regardless of their chemical characteristics, but only depending on their saturation concentration. Therefore it becomes possible to add aging processes in the simulation of OA by adding reactions that shift species from one volatility bin to another (Donahue et al., 2006). This scheme was implemented and tested in CHIMERE for Mexico city (Hodzic and Jimenez, 2011) and the Paris region (Zhang et al., 2013). The volatility profile used for this scheme consists of nine volatility bins with saturation concentrations in the range of 0.01 to $10^6$ μg.m$^{-3}$ (convertible to saturation pressure using



atmospheric standard conditions), across which the emissions of SVOC and IVOC (Intermediate Volatility Organic Compounds) are distributed, following a specific aggregation table (Robinson et al., 2007). Four volatility bins are added for anthropogenic and biogenic SOA ranging from 1 to 1000 µg.m$^{-3}$. SOA yields are taken from the literature (Murphy and Pandis, 2009, Lane et al., 2008) using low-NO$_x$ condition (VOC/NO$_x$ > 10ppbC.ppb$^{-1}$).

The SVOC species can age, by decreasing their volatility by one bin independent of their origin (e.g directly emitted or formed from anthropogenic or biogenic VOC precursors) with a given constant rate. Fragmentation processes and the production of non-volatile SOAs are ignored in this scheme. In the basic VBS scheme, the BSOA aging processes are usually ignored since they tend to result in a significant overestimation in biogenic SOA (Lane et al., 2008). Although physically present, their kinetic constants for this aging process are considered the same as

anthropogenic compounds and seem to be overestimated. However, in Zhang et al. (2013), including BSOA aging was necessary to explain the observed experimental data. Therefore, in this work, the VBS scheme is evaluated with and without including the BSOA aging processes. Figure 2-b shows a simplified illustration for anthropogenic and biogenic SOA, while the partition for SVOC is presented in supplementary information (SI2). For all bins, regardless of their origin, the partitioning between gaseous and the particulate phases is performed following

Raoult's law and depends on total organic aerosol concentration. Under normal atmospheric conditions, SVOC with the volatility range of 0.01 to 10$^3$ µg.m$^{-3}$ can form aerosols. In total, the sum of 24 species in the model (with 10 size distribution bins each, i.e. 240 species in total) makes up the total concentration of SOA simulated by this scheme.

**2.2.3 Modified VBS scheme**

The basic VBS scheme does not include fragmentation processes, corresponding to the break-up of oxidized OA compounds in the atmosphere into smaller and thus more volatile molecules (Shrivastava et al., 2011). It also does not include the formation of non-volatile SOA, where SOA can become non-volatile after formation (Shrivastava et al., 2015). In this work, these two processes were added to the VBS scheme presented above and tested for the Mediterranean basin. The volatility bins for the VBS model were not changed (ranges presented in the previous

section). SOA yields were kept as in the standard VBS scheme, however, instead of using the low-NO$_x$ or the high-NO$_x$ regimes, an interpolation between the yields of these two regimes was added to the model. For this purpose, a parameter is added to the scheme, which calculates the ratio of the reaction rate of RO$_2$ radicals with NO (high-NO$_x$ regime) with respect to the sum of reaction rates of the reactions with HO$_2$, and RO$_2$ (low-NO$_x$ regime). The fragmentation processes for the SVOC start after the third generation of oxidation, because fragmentation is

favored with respect to functionalization for more oxidized compounds. Therefore, three series of species in different volatility bins were added to present each generation, similar to the approach set up in Shrivastava et al. (2013). However for biogenic VOC, fragmentation processes come into effect starting from the first generation, as in Shrivastava et al. 2013, because the intermediate species are considered as more oxidized. A fragmentation rate of 75% (with 25% left for functionalization) is used in this work for each oxidation step following Shrivastava

et al. (2015). The formation of non-volatile SOA is performed by moving a part of each aerosol bin to non-volatile bins with a reaction constant of corresponding to a life time of one hour, similar to Shrivastava et al. (2015). Figure 2-c shows a scheme of the modified VBS for the VOC. In total, 40 species (with 10 size distribution bins each, i.e. 400 species in total) are added together to calculate the total concentration of SOA simulated by this scheme. The resulting model has a total of 740 species in the output files (including gas-phase chemistry), which makes this





scheme the most time consuming among the tested schemes. In section 5 of this paper, the results from the three schemes introduced above will be compared to observations.

## 3 Experimental dataset

During the SAFMED subproject, measurements were made at two major sites, the Ersa, Cap Corse station and the
Cap Es Pinar, Mallorca station. The geographical characteristics and the measurements performed at each site are presented in the following section.

### 3.1 ChArMEx measurements

The Ersa supersite (42°58'04.1", 9°22'49.1") is located on the northern edge of the Corsica Island, in a rural environment, at an altitude of 530m above sea level (asl). The station is located on a crest that dominates the
northern part of the cape. It has a direct view of the sea on the three western, northern and eastern sides. Measurements carried out in this station are reported in table 2. More details about the instrumental setup in Ersa can be found in Michoud et al. (2017), Arndt et al. (2017) and Pey et al. (in prep).

The Es Pinar supersite (39°53'04.6", 3°11'40.9") is located on the northeastern part of Mallorca. The monitoring station was placed in the "Es Pinar" military facilities belonging to the Spanish Ministry of Defense. The
environment is a non-urbanized area surrounded by pine forested slopes, being one of the most insulated zones in the Mallorca Isle, in between the Alcudia and Pollença bays, but still can be influenced by local anthropogenic emissions. The site is located in an altitude of 20 m.asl. The location of the station and a list of available measurements are presented in figure 3 and table 2 respectively.

For both sites, NOx (nitrogen oxides) were measured by CraNOx analyzer using ozone chemiluminescence with
a resolution of 5 minutes. A photolytic converter was incorporated in the analyzer to convert direct measurements of NO to $NO_2$. Volatile Organic compounds were monitored at both supersites by a PTR-ToF-MS (Kore 2nd generation at Ersa, and Ionikon PTR-ToF-MS 8000 at Es Pinar). A detailed procedure of VOCs quantification is provided in Michoud et al (2017) for Ersa and Pey et al (in prep) for Es Pinar. Briefly, both instruments were calibrated daily using gas standard calibration bottles and blanks performed by means of a catalytic converter
(stainless steel tubing filled with Pt wool held at 350°C).

A Quadrupole Aerosol Chemical Speciation Monitor (Q-ACSM, Aerodyne Research Inc, Ng et al., 2011) was used for the measurements of the chemical composition of non-refractory submicron aerosol at Ersa. This instrument has the same general structure of AMS (Aerosol Mass Spectrometer), with the difference that it was developed specifically for long term monitoring. A High-Resolution Time-of-Flight AMS (HR-ToF-AMS,
Aerodyne Research Inc, Decarlo et al., 2006) operated under standard conditions (i.e. temperature of the vaporizer set at 600 °C, electronic ionization (EI) at 70 eV) was deployed with a temporal resolution of 8 minutes to determine the bulk chemical composition of the non-refractory fraction of the aerosol and for the Es Pinar site. AMS data were processed and analyzed using HR-ToF-AMS Analysis software SQUIRREL (SeQUential Igor data RetRiEvaL) v.1.52L and PIKA (Peak Integration by Key Analysis) v.1.11L for IGOR Pro software package
(Wavemetrics, Inc., Portland, OR, USA). Q-ACSM and AMS source apportionment results discussed in this work are detailed in Michoud et al. (2017) and Pey et al. (in prep) respectively. For both sites, source contributions were obtained from PMF analysis (Paatero and Tapper, 1994) of Q-ACSM and AMS spectra. PMF was solved using



the multi-linear engine (ME-2, Paatero, 1997) algorithm, using the Source Finder toolkit (SoFi, Canonaco et al., 2013). For the Ersa site, HOA (Hydrocarbon-like Organic Aerosol) profile was constrained to a value of 0.1. Two other factors were extracted including SVOOA (Semi-Volatile Oxygenated Organic Aerosol) and LVOOA (Low-Volatile Oxygenated Organic Aerosol). For the Cap Es Pinar site, because of its very low contribution in such

remote location, HOA has been constrained using an a-value of 0.05. In addition to HOA, 3 other factors have been extracted from the PMF analysis: SVOOA (Semi-Volatile Oxygenated Organic Aerosol) and 2 LVOOA (Low Volatile Oxygenated Organic Aerosol) factors. Differences between these 2 LVOOA factors are mainly linked to air masses origins and to the probable influence of marine emissions for one of the LVOOA factor at both sites, with correlation coefficient ($R^2$) of 0.43 and 0.47 with the main fragment derived from methane sulfonic

acid (MSA, fragment $CH_3SO_2^+$) for Es Pinar and Ersa, respectively. For a sake of clarity and for intercomparison purpose with the model outcomes, we merge the 2 LVOOA factors into one LVOOA for Cap Es Pinar site to be compared to the Ersa site results. Online aerosol chemical characterization was complemented by an Aethalometer (AE33, MAGEE, Drinovec et al., 2015) at Es Pinar and a Multiangle Absorption Photometer (MAAP5012, Thermo) for the quantification of BC. $PM_{10}$ total mass measurements were taken from TEOM-FDMS (Tapered

Element Oscillating Microbalance-Filter Dynamic Measurement System) for Es Pinar and BETA corrected by factor obtained after comparison to gravimetric for Es Pinar. Finally, daily $PM_1$ aerosol samples were collected onto 150 mm quartz fiber filters (Tissuquartz, Pall). Eighteen and eight samples were selected for Ersa and Es Pinar, respectively, for a subsequent analysis of radiocarbon performed on both OC and EC fraction following the method developed in Zhang et al. (2012).

**3.2 Other measurements**

For the validation of meteorological parameters, along with the measurements in ChArMEx stations, other datasets were also used. Radio-sounding data for three stations in the western Mediterranean basin were used for simulation-observation comparisons for meteorological parameters. The radio-soundings are performed by Météo-France at the two stations of Ajaccio, France (41°55'5", 8°47'38") and Nîmes, France (43°51'22", 4°24'22") and

by AEMet at Palma, Spain (39°36'21", 2°42'24"). Each day two balloons at about 00 and 12 UTC are available for each station, a total of 96 balloons are included in the comparisons. Ajaccio and Palma are coastal stations, but Nîmes is farther from the coast compared to the other two stations.

**4 Model validation**

The CHIMERE model has been previously validated for different parts of the world (Hodzic & Jimenez, 2011 ;

Solazzo et al., 2012 ; Borrego et al, 2013 ; Berezin et al., 2013 ; Petetin et al., 2014 ; Rea et al., 2015 ; ; Konovalov et al., 2015 ; Mallet et al., 2016). The data set presented in section 3 is used for model validation. First, a representativeness error within simulations is calculated for a list of pollutants, which is necessary to distinguish between uncertainties due to limitations in model resolution and due to other reasons. Then a validation for the meteorological parameters is presented, before comparison of gaseous and aerosol measurements.





### 4.1 Orographic representativeness of Cap Corse simulations

As explained before, during the ChArMEx campaign, an important number of observations were made at Ersa, Cap Corse. In order to use this data set for model evaluation, potential discrepancies due to a crude representation of the complex orography of Cap Corse need to be minimized and quantified since the measurements were

performed on the crest line.

For the 10-km domain (D10), we noticed that there was an inconsistency between simulated and real altitude of the Ersa site; altitude being simulated at 360 m asl lower than the real altitude of measurements (530 m asl). Therefore, 1-km horizontally-resolved simulations were performed for the domain. However, even for the 1-km simulations the simulated altitude is too low (365 m asl). This error occurs because the altitude of each cell in

CHIMERE is calculated using the average of altitudes of points inside the cell, therefore if the altitude of the ground surface inside a cell happens to vary greatly, the average would be lower than the higher points seen in the cell (which corresponds in our case to the Ersa site located on the crest). In addition, the average of the marine boundary layer height is typically around 500 m (Stull, 1988), therefore a discrepancy in the simulated altitude could cause significant errors in the simulations. These two facts, make it important to explore the

representativeness of the simulations regarding this station.

This led us to perform an orographic representativeness test on the 1-km domain (D1) at the Ersa site. A matrix of neighboring cells around the grid cell covering the Ersa station (up to 5km distance) was taken (figure 4-a), and species concentrations were plotted against the variation of the altitude of these different cells. The highest altitude reached by one of the cells is about 450 m. Then, the concentration on the exact altitude (530 m) was extrapolated

using a non-linear regression between the altitude and the concentration of the selected cells with several different equations for each time step. In total, 9 non-linear equations were tested, from which only five were finally used for the calculation of the representativeness error due to convergence problems for the other four equations (see supplementary material, SI3 for the details). Regressions were performed separately for each of the 720 hourly time-steps of the one month simulations. An example of this regression for organic aerosols for one time step and

one of the equations is shown in figure 4-b (equation n.1 from SI3). The results were filtered using two criteria (convergence of regression for each time-step and a correlation coefficient between fitted and simulated points of higher than a fixed value) depending on statistical values of the regressions (see supplementary material, SI3, for details) and only regressions conforming to these criteria were retained. If at least two converging regressions were not retained for a given time step, the results for that time step were not further used. Figure 4-c shows the compiled

results for all equations and all simulation times in one time series for total OA concentration. Note that model output was generated with an hourly time step. Using these results, for a list of different species, an orographic representativeness error (ORE) was calculated using the average of the difference of the upper and lower confidence intervals for all equations. As an example, carbon monoxide, which is a well-mixed and a more stable component in the atmosphere, presents the lowest error among the tested species (2%). Ozone also presents one

of the lowest errors (4%) and nitrogen oxides one of the highest (75%). Organic aerosol, of particular interest for this study, shows a moderate error of 10%. In terms of meteorological parameters, relative humidity appears more affected (relative ORE of 18%) than temperature (the relative ORE is calculated on values of T in °C). A summary of results of this test is shown in table 3.

A general conclusion is that secondary pollutants appear to be well represented from a geographic point of view,

however model-observation comparisons for specially more variable primary pollutants should be performed with



caution and keeping in mind the fact that the simulated altitude is not representative of the orography for this specific station. This is due to the fact that for short-lived primary species, if emission sources happen to be nearby (which is the case here), then the air masses in the area have not had yet the chance to properly mix. On the contrary, secondary species, which partly have been transported from the continental boundary layer, due to their

lifetime of several days to several weeks, are believed to be better mixed. Therefore, for secondary species differences in the simulated versus observed altitude lead to relatively smaller errors.

The question remains on which domain should be used for model-measurement comparisons remains. As seen above, D10, despite having a sufficiently fine resolution for most continental areas, is not capable of representing the complex orography of the Cap Corse, therefore the D1 simulation results have been used for comparisons,

except for meteorological parameters, where all possible domains are compared. The Es Pinar station does not have the same intense altitudinal gradient that is seen at Ersa, therefore the aforementioned test was not performed for this station and the D10 simulations are used for comparisons.

In the following section, for each model-observation comparison at Ersa, a confidence interval which stands for the ORE value that was derived in this section is added.

**4.2 Meteorology evaluation**

Meteorological output of the mesoscale WRF model at different resolutions has been used as input to the CHIMERE CTM. The meteorological data used by CHIMERE was compared to various meteorological observations such as radio-soundings and surface observations at the measurement sites. Detailed results of these comparisons are given in the supplementary material (SI4). Here, a short overview of the results and the

implications for the model ability to simulate transport to the measurement sites is given.

Comparisons for temperature, a basic variable to control the quality of meteorological simulations show a good correlation for radio-sounding comparisons (typically from 0.60 to 0.85 for hourly values) and a low bias (typically from -1.16 to -0.39°C) for three sites of Palma, Nîmes and Ajaccio. Wind speed shows a good correlation at higher altitudes, and also near the surface for Nimes and Palma stations for radio-sounding stations, while for the Ajaccio

station the sea/land breezes are probably not well represented in the model. For Es Pinar, the coastal feature of the site is difficult to take into account in a 10 km horizontally resolved simulation and leads to larger errors. Ground based measurements were also compared (SI4), whereas at Ersa, the correlation in finer domains is better than that of D10 for wind speed (typically around 0.66 versus around 0.60). For the E-OBS network (Datasets provided by European Climate Assessment & Dataset project for monitoring and analyzing climate extremes, Haylock et al.,

2008; Hofstra et al., 2009) comparisons (SI4) also show a good correlation and a low bias for temperature (correlation of 0.79 with a bias of -0.54°C for mean temperature observed for 71 stations in D10), while the daily minima seems to be underestimated (bias of -3°C for daily minima observed for 71 stations).

While the general comparison between the meteorological fields used as input for CHIMERE simulations and observations is in general satisfying, when daily averages representative for regimes are compared instead of

hourly values, the correlation becomes higher and the bias lower. Differences in wind speed and wind direction may lead to errors in representing the short term variability of gaseous and aerosol species.



### 4.3 Gaseous species

Among all the gaseous species available in the observations, four were chosen in this study: nitrogen oxides ($NO_x$), isoprene ($C_5H_8$), monoterpenes and the sum of methacrolein and methyl vinyl ketone (here after called MACR+MVK). These four species were chosen since isoprene and monoterpenes are the principal precursors for biogenic SOA, MACR+MVK are formed during isoprene oxidation and $NO_x$ is a good tracer for local pollution. The comparisons for the Ersa and Es Pinar stations are shown in figure 5, and statistics of the comparison are shown in table 4. For Ersa, the orographic representativeness errors derived in section 4.1 are also shown. In all comparisons, the results for the simulations with the modified VBS scheme are used, but the choice of the organic aerosol scheme only slightly affects the simulation of gaseous species (mainly via heterogeneous reactions on aerosol surfaces included within CHIMERE).

For nitrogen oxides (figure 5-a1), results show that there is a good correspondence between the averages of simulated and observed nitrogen oxides at Ersa. The low correlation for nitrogen oxides at Ersa might be partly explained by the high representativeness error (75%) for this component. This is because the altitude in the simulations is lower and therefore the emission sources are closer in the model than they are in reality. At Es Pinar (figure 5-a2), both observed and simulated NOx concentrations becomes higher than at Ersa, indicating the more locally polluted nature of this site, but the model underestimates observed NOx by about 40%.

For isoprene (figure 5-b1 and 5-b2), a good correlation (0.76, 0.71) between simulations and observations appears at both sites. However there is an important overestimation (by a factor of 2.5) in the simulations for the Ersa site, which could also be linked to the high orographic representativeness error (85%), and also to the fact that local emissions sources are not correctly taken into account in the MEGAN emission model. On the contrary, at Es Pinar isoprene is underestimated by about 25%. The sum of MACR+MVK (figure 5-c1 and 5-c2) is overestimated by about a factor of two at Ersa, following the pattern of overestimation of isoprene at this site, while the bias is small at Es Pinar. Monoterpenes (figure 5-d1 and 5-d2) show an underestimation by about 70% at both sites, observations being about 5 times larger at Ersa than at Es Pinar. Again, this could be related to the orographic representativeness error at Ersa and to in-accounted for local vegetation at both sites.

Daily correlations of 0.35, 0.87, 0.85 and 0.58 (instead of 0.37, 0.76, 0.62 and 0.35 hourly values) are seen at Ersa for nitrogen oxides, isoprene, MACR+MVK and monoterpenes, respectively. These values change to 0.16, 0.51, 0.72 and 0.10 for daily comparisons (instead of 0.12, 0.69, 0.41 and 0.14 when correlating hourly values) at Es Pinar. While showing the same improvement in correlation as for meteorological parameters for Ersa, meaning that chemical regime changes are better simulated than short term variations, the results do not change as much for Es Pinar.

Isoprene and monoterpenes are the most important precursors of biogenic organic aerosols. Sensitivity tests using the modified VBS scheme showed that there is a 40 to 60% repartition for the SOA formed from isoprene and monoterpenes, respectively, in the western Mediterranean region. A drawback of the comparisons for these two species is the fact that the measurements present a very local point of view, while a more regional look would have been more pertinent for these species, since a significant amount of SOA is transported to both sites mainly from continental areas.

As mentioned before (section 2), isoprene and terpene emissions in our work are generated by the MEGAN model (Guenther et al., 2006). Zare et al. (2012) evaluated MEGAN model derived isoprene emissions coupled to the hemispheric DEHM chemistry-transport model against measurements. On average over 2006, they found a

simulated isoprene overestimation for 4 European sites (between a factor 2 and 10), good agreement (within +/-30%) for 2 sites and an underestimation for 2 sites (also between a factor 2 and 10). However, none of the sites was located close to the Mediterranean Sea. Curci et al. (2010) performed an inverse modelling study to correct European summer (May to September) 2005 MEGAN isoprene emissions from formaldehyde vertical column OMI measurements (given that isoprene is a major formaldehyde precursor). For western Mediterranean countries area, they found an isoprene emissions underestimation using MEGAN of 40 % over Spain, a tendency for an underestimation, but with regional differences over Italy, and only small differences over France. This comparison globally lends confidence to MEGAN derived isoprene emissions. Unfortunately, to our best knowledge, no comparable studies exist in order to validate monoterpene emissions. For the Rome area, Italy, Fares et al. (2013) concluded for a variety of typical Mediterranean tree and vegetation species mix, that MEGAN correctly simulated mean observed monoterpene fluxes over the last two weeks of August 2007 within 10% error, as long as a canopy model was included (as in our study).

**4.4 Particulate species**

Figure 6 shows the comparison between observations and simulations for particulate sulfate and black carbon in $PM_1$. These two species are chosen as two important fine aerosol components, before the comparison of organic aerosol in chapter 5. The left panel shows the comparison for Ersa and the right one for Es Pinar. Statistical information for these species is given in table 5. There is an overestimation for sulfate particles (figures 6-b1 and 6-b2) by about 45%, well beyond the representativeness error for this species (15%). Besides, the short and sharp decreases in the measurements of sulfates correspond to low clouds passing at the level of the station which are not simulated by the model. A large sulfate peak simulated in the morning of 29 July is not present in the observations; it originates in the model from an air mass arriving from Marseille which is both a busy harbor and an important industrial area with large $SO_2$ emissions. This very specific transport event is not observed, which is probably due to small errors in the wind fields. In addition, there are two periods of overestimation of this species in the simulations; the period of 18 to 20 July and the period of 29 July to the night of first of August. During the second period, the ACSM $PM_1$ observations show concentrations of close to zero, which are consistent with $PM_{10}$ PILS-IC sulfate measurements. For this period elevated southerly winds are observed in the Corsica area, and the absence of strong $SO_2$ sources in this sector might explain the lower concentrations that are seen not only for sulfate but also for black carbon. The constant overestimation of sulfates during this period may suggest an overestimation of boundary conditions for these species.

A moderate correlation (0.26) between the model and the observations is seen for BC (figures 6-a1 and 6-a2), but the representativeness error is important for this species (26%), since one of its primary origins is emission of ships passing nearby the coasts of the Cap Corse. The local emissions due to shipping activities at Ersa and anthropogenic activities at Es Pinar are visible in the observations as frequent narrow (in time) peaks. These activities are also visible in the simulations at Ersa. However, at Es pinar the model doesn't succeed in simulating them, as already observed to an even larger extent for NOx. The slight overestimation of BC (5%) in the model for the Ersa site could be explained by the orographic representativeness error. The same cloud effect seen for the sulfate particles is visible for BC for the Ersa site as well. Correlation between daily observations and the simulations was also calculated which shows 0.51 and 0.56 for the Ersa station (instead of 0.36 and 0.42) and 0.81 and 0.65 for the Es pinar station (instead of 0.47 and 0.52) for sulfate and BC respectively. Similar to





meteorological parameters, the model can reproduce the daily concentration changes for these two species better than the hourly changes.

As a conclusion of this chapter, the comparison between the meteorological fields used as input for CHIMERE simulations and observations at surface sites and from radio-soundings is satisfying, and shows low bias. However, differences in wind speed and wind direction may suggest that the variability in advection to the measurement sites, in particular at short timescales (less than a day), is not always properly simulated. This may lead to errors in representing the short-term variability of gaseous species. Among gaseous species, the biases found for the short-lived compounds could mainly be explained by strong orographic representativeness error, which were evaluated in the range 60-85% in this section. The local character of the comparisons for biogenic VOCs was pointed out, as well as the fact that biogenic OA is formed at a regional scale from these precursors rather than a local scale. Available regional comparisons show acceptable comparisons for BVOC observed and simulated concentrations (Curci et al., 2010 ; Zare et al., 2012 ; Fares et al., 2013).Therefore, a systematic misrepresentation of BVOC in the model seem to be unlikely. For aerosol species, low bias was seen for BC, while an overestimation in simulations is observed for sulfates.

## 5 Organic aerosol simulation

A description of each of the 4 schemes for the simulation of organic aerosols tested within the CHIMERE model for the Mediterranean area has been given in section 2. These four schemes are the CHIMERE standard scheme, the VBS scheme with BSOA aging (noted as Standard VBS_ba), the VBS scheme without BSOA aging (noted as Standard VBS_nba) and the modified VBS (noted as modified VBS) scheme which includes fragmentation and formation of non-volatile organic aerosol. For each scheme, four domains were used: one coarse domain, and three others nested inside the coarse one with increasing resolutions. As before, the simulations from the finest domain are used for the comparisons. The domain with the finest resolution (1km) was used for the representativeness tests. For each scheme, meteorological and boundary conditions are the same, hence only the simulation of organic aerosols and subsequently the aggregation of anthropogenic PM emissions differ between simulations.

### 5.1 Comparison of PM$_1$ total organic aerosols concentration

Figure 7 shows the time series of comparison of organic aerosols for these four schemes with the measurements at Ersa and Es Pinar. The circles beside each plot show the average concentration for different time series, in addition table 6 shows the statistic parameters corresponding to these time series. The observed OA concentration at the Ersa site measured by ACSM for the PM$_1$ fraction has an average of 3.71 µg.m$^{-3}$ and that of the Es Pinar site measured by AMS for the PM$_1$ fraction a somewhat lower average of 2.88 µg.m$^{-3}$. The difference between the two sites may be attributed to the fact that Cap Corse is closer to both local and transported (continental) biogenic sources than Es Pinar. The VBS scheme with BSOA aging greatly overestimates (bias of more than a factor of three) the organic aerosol; as mentioned before, the aging of biogenic aerosols in the VBS scheme usually results in an overestimation in organic aerosols (Lane et al., 2008). With the BSOA aging option turned off, the standard VBS scheme comes much closer to the average of total OA concentration measured at both sites (relative biases below 50%). The CHIMERE standard scheme also overestimates the mass concentration of organic aerosols, but



in a lesser extent compared to the VBS scheme with BSOA aging (bias of a bit less than a factor of two). The modified VBS scheme corresponds much better to observations (negative biases about 20%). The model results agree for both stations in this regard. At Ersa, biases of +244%, +91%, +34% and -18% respectively are observed for the VBS standard scheme with BSOA aging, the CHIMERE scheme, the standard VBS scheme without BSOA

aging and the VBS modified scheme respectively, the corresponding numbers are of +218%, +98%, +43% and -23% for Es Pinar.

The daily correlation for the modified VBS scheme is 0.63 and 0.51 for Ersa and Es Pinar respectively (instead of 0.50 and 0.29 for hourly values). This shows, again, that the model represents long-scale changes better than hourly variations. While the concentrations of both the modified VBS scheme and the standard VBS scheme without

BSOA aging correspond well with the experimental data, other aspects such as origins of the formed organic aerosol and its oxidation state have to be considered before reaching any conclusion about the robustness of these schemes. It is also noticeable that the large-scale tendencies are almost respected in each scheme, therefore different regimes are well predicted by the model. However, smaller changes in the observations (on shorter time scales, a day or less) are not reproduced in any of the schemes. This might be because of wind direction simulation,

which, as explained before is difficult to represent in coastal areas.

The overestimation of secondary organic aerosol with the CHIMERE standard scheme is a new feature (Bessagnet et al., 2008 ; Hodzic and Jimenez, 2011 ; Petetin et al., 2014). A previous comparison of CHIMERE with organic aerosol simulations using the same scheme and coupled to MEGAN biogenic emissions, resulted in a good comparison or underestimation with OC observations over Europe from the Carbosol project (Gelencsér et al.,

2007) for summer 2003, when biogenic SOA was dominant. However, this comparison included only one site close to the western Mediterranean basin (Montelibretti, Italy). The overestimation of BSOA in the VBS scheme when the BSOA aging is activated was documented for US in several occasions (Robinson et al., 2007b ; Lane et al., 2008). Ultimately, it cannot be excluded that the OA overestimation with both schemes at Ersa and Es Pinar is also due to BSOA overestimation. However, the available material does not support this hypothesis: (i) OA

overestimation is observed at two independent, distant sites, (ii) local monoterpene underestimation at both these sites, (iii) no evidence for MEGAN monoterpene overestimation is available in literature and MEGAN isoprene overestimation over the western Mediterranean area was ruled out by satellite observations (Curci et al., 2010).

### 5.2 Total carbonaceous particles origins based on [14]C measurements

The results of [14]C measurements in carbonaceous aerosol filter samples for the PM$_1$ fraction at the Ersa and Es

Pinar sites were used in order to better discriminate between the modern and mostly biogenic versus fossil and anthropogenic origin of organic aerosols, and to compare it to simulations with different organic aerosol schemes. It must be noted that in the simulations, species are not separated automatically into fossil/ non-fossil parts, therefore these fractions need to be calculated as a post-treatment of simulations, affecting each relevant particulate species to both fractions (Table 7). For carbonaceous aerosol, residential/domestic uses are considered as non-

fossil as they are mostly related to wood burning (Sasser et al., 2012), therefore, they are attributed to the non-fossil bin (3.6% for BC and 12.3% for OC, Sasser et al., 2012). The non-fossil contribution of ASOA and POA due to bioethanol usage is ignored here (<5%). No major biomass burning events were seen in the period of this study, but minor contributions of this source cannot be excluded. It should also be mentioned that the [14]C measurements show the mass of carbonaceous particles in each filter, therefore have a unit of µgC.m$^{-3}$, while the



simulations show the total organic aerosols in µg.m$^3$. In the comparisons that follow, it is pertinent to use an OM/OC conversion factor to be able to compare the $^{14}$C measurements to simulations. For this purpose, an average OM/OC factor of 2 is used for the secondary aerosol fraction (both for the LV-OOA and SV-OOA factors), while a factor of 1.3 is used for HOA according to Aiken et al.(2008). However, the choice of the OM/OC factor has a small effect on the outcome of general comparisons since the comparisons are all shown in percentage of fossil/non-fossil distribution.

Figure 8-a for Ersa and 8-b for Es Pinar show the average of all filters for each scheme compared to the observations, while figure 9-a and 9-b show the relative distribution of fossil/non fossil sources at both sites. Among these total averages, the distribution at Ersa is 81% non-fossil and 19% fossil, and 67% versus 33% at Es Pinar. Apparently, biogenic contributions to OA are dominant at both sites, but larger for the Ersa site.

While the comparison of averages for all schemes with Ersa measurements shows that the modified VBS scheme is the closest in fossil/non-fossil partitioning (19%/81%), the CHIMERE standard scheme is also performing well when looking at the percentage of distribution for each source (20%/80%). The VBS scheme with BSOA aging shows an underestimation in the percentage of fossil carbons (16%/84%), which can be due to the overestimation of biogenic aging of secondary organic aerosols in this scheme; on the contrary, the VBS scheme without BSOA aging shows an important overestimation of the fossil percentage (34%/66%).

A distribution of 33%/67% of the fossil-non-fossil fraction is observed at Es Pinar. As for the Ersa site, the modified VBS scheme is the closest to the observations (32%/68%). The CHIMERE standard scheme shows an underestimation of the fossil contribution with a distribution of 28%/72%. The standard VBS scheme with BSOA aging underestimates the fossil contribution with a distribution of 21%/79%, while, the standard VBS scheme without BSOA aging largely overestimates the fossil contribution (42%/58%). These results show that the two sites differ greatly when taking into account nearby sources and geographical conditions; the Ersa site is an elevated rural station at the interface between the marine boundary layer and the residual boundary layer, and the Es Pinar site is a sea-side station closer to anthropogenic sources. These differences should normally be represented in the percentage of fossil carbon concentrations both in observations and in simulations. Although this difference is noticeable in observations, and also in the modified VBS scheme, it is less emphasized in the CHIMERE standard scheme and in the VBS standard scheme with either parameterization of aging.

A more detailed look to individual filters for the modified VBS scheme for the Ersa and Es Pinar sites is presented in figure 9-a and figure 9-b respectively. The tendencies from one day to another are in most cases not reproduced correctly for both sites. Therefore while the average mass repartition of modern and fossil sources is well-represented by this scheme, the day-to-day variability is not fully consistent with the measurements. This inconsistency is also seen in the other tested schemes.

**5.3 Volatility and oxidation state comparison with PMF results**

The PMF (Positive Matrix Factorization) results of the ACSM/AMS (Aerosol Mass Spectrometer) measurements give us the chance to learn more about the oxidation state of the organic aerosols (Michoud et al., 2017 for Ersa and Pey et al. in prep for Es Pinar). PMF analysis allows to divide PM$_1$ organic aerosol measurements into different groups with distinctive mass spectra corresponding to distinctive oxidation state (Lanz et al., 2010). The most common retrieved factors of such an analysis are HOA (Hydrogen-like Organic Aerosol), SV-OOA (Semi-



Volatility Oxidized Organic Aerosol) and LV-OOA (Low-Volatility Oxidized Organic Aerosol). However, it does not give direct information about the volatility range of each group. This makes the PMF output difficult to compare with our model results, which give the volatility distribution of organic aerosol, but not its oxidative state. Here, we first compare volatility distributions obtained with the four aerosol schemes, and then try to attribute the

simulated aerosol to the three factors HOA, SV-OOA and LV-OOA, in order to compare it to the observed distributions. The three schemes based on the VBS scheme already distribute aerosols in volatility bins (Robinson et al., 2007a). The distribution to LV-OOA and SV-OOA for these three schemes was done mainly by taking into account the saturation concentration, with the threshold chosen as a saturation concentration $C* \geq 1$ µg.m$^{-3}$ for SV-OOA and $C* \leq 0.1$ µg.m$^{-3}$ for LV-OOA (Donahue et al., 2012). Primary organic aerosols were considered to

be in the HOA factor regardless of their saturation concentration. For the CHIMERE standard scheme, each surrogate species is associated with a saturation vapor pressure, which was used to calculate the saturation concentration for each component at ambient temperature.

The observations show that at Ersa, the LV-OOA factor dominates the PMF results (88%), with 10% SV-OOA and a minor (only 2%) contribution of HOA. For the Es Pinar site the contribution of LV-OOA drops to (75%)

and the contribution of HOA and SVOOA becomes somewhat larger (4% and 21% respectively). As mentioned before, the Es Pinar site is more influenced by local anthropogenic sources. Therefore, more local OA emissions are expected which corresponds to the increase in the HOA percentage seen in the observations. These emissions are oxidized locally to form SVOCs that fall in the SV-OOA group explaining the rise in the percentage of this group.

Figure 10-a and figure 10-b show the relative distribution of all organic aerosols in seven volatility bins for all tested schemes for Ersa and Es Pinar sites, respectively, and for all tested schemes. For each scheme, the average for the total period of the simulations was used to calculate the percentage in each bin. The bins shown are in the range of $10^{-3} – 10^3$ µg.m$^{-3}$, all the aerosols with a volatility higher or lower than the extremes are put in the last high or low bin respectively.

These figures show that the percentage of aerosols in each volatility bin for the two different sites is relatively similar. The CHIMERE standard scheme distributes most of the aerosol in the three bins in 1-100 µg.m$^{-3}$ volatility ranges which falls into the SV-OOA group obtained by PMF. This is due to the initial distribution of volatilities of surrogate SVOC species used that best fit chamber measurements, and the fact that there is no aging mechanism in this scheme to make the aerosols less volatile. The 100µg.m$^{-3}$ volatility bin corresponds to SVOCs produced

from isoprene and monoterpene oxidation and presents the highest percentage for this scheme. The standard scheme also produces 16% of organic aerosols in the LV-OOA range (volatilities corresponding to a $C*$ between 0.001 and 0.1 µg.m$^{-3}$). The standard VBS scheme with BSOA aging has only small fraction of aerosol in the LV-OOA range (from SVOC emission aging) since, by construction of the scheme, the most aged BSOA and ASOA aerosols fall in the 1µg.m$^{-3}$ volatility bin, which is actually the bin with the highest percentage of OA. The standard

VBS scheme without BSOA aging presents relatively similar results to the scheme with biogenic aging, with as expected larger percentages for higher volatilities in the absence of aging, It also show a larger LV-OOA fraction, probably because the lower total OA concentration favors the contribution of lower volatility SVOC's to the aerosol phase. On the whole, the two schemes yield much too low LV-OOA fractions as compared to observations. The modified VBS scheme has a more realistic distribution into the three oxidation groups. In this scheme, the

highest percentage of OA falls in the $10^{-3}$ µg.m$^{-3}$ volatility bin which is in the LV-OOA range. The rest of the





aerosols are distributed almost equally in volatility bins between $10^{-2}$ and $10^{2}$ µg.m$^{-3}$ with a slight decrease in percentage in the higher volatility bins. The percentage in higher volatility bins at Ersa are slightly lower than the ones at Es Pinar which could be explained by the stronger local sources at the Es Pinar site with stronger primary SVOC emissions.

Figure 11-a and figure 11-b show the calculated HOA, LV-OOA and SV-OOA groups in simulations compared to the ones in observations at Ersa and at Es Pinar respectively. For the Ersa site the results of the modified VBS scheme are consistent with the observations with a slight underestimation of HOAs which is more visible at Es Pinar. The standard CHIMERE scheme leads to higher, overestimated HOA values, as primary OA emissions are considered non-volatile. Only the modified VBS scheme succeeds to reproduce the major contribution of LV-

OOAs at both sites. However, while staying close to observations at Ersa, it seems to overestimate the formation of LV-OOAs at Es Pinar. The standard VBS schemes with or without BSOA aging, greatly underestimate the formation of LV-OOAs, and as a counterpart overestimate the formation of SV-OOAs. Therefore, among the tested schemes, only the modified VBS scheme allows representing the distribution of the different PMF factors in a satisfying way.

A general conclusion for this section is that among the tested schemes, the modified VBS scheme seems to be closer to observations in all three criteria chosen for this study (mass, modern vs. fossil origin, volatility/oxidation state). It succeeds to reproduce the mass average of the aerosols observed at both sites, represents the fossil/non-fossil distribution of OA both when looking at the percentages and simulated concentrations, and it comes closest to observations when comparing the oxidation state.

**6 Budget of organic aerosols**

In section 5, we have highlighted the best performance of the modified VBS scheme for organic aerosol simulation amongst the tested schemes by comparison with observational data at two different sites in the western Mediterranean basin. In the present section, we use this scheme in order to simulate the organic aerosol distribution and its anthropogenic and biogenic origin over the western Mediterranean basin during the SAFMED campaign.

Figure 12 shows a series of figures where the left column always corresponds to simulations near the surface, and the right column shows the same concentration at an altitude of between 300 to 450m (for marine grid cells, called for simplicity boundary layer BL). Each row shows a different component, first row corresponding to OA concentrations, second row to biogenic OA concentrations, third row to anthropogenic OA concentrations and last row presents the sum of POA and all its subsequent oxidation products. The bigger part of each figure corresponds

to D10 simulations, while the part inside the black rectangle shows the D3 simulations, both showing the average of the organic aerosols on the whole simulation period (1 month form July 10$^{th}$ to August 9$^{th}$). Figure 13 shows the same type of figures for the percentage of contribution of biogenic OAs, anthropogenic OAs and the sum of POAs (that is POA and POA oxidation products). The small differences at the interface of the two domains is because CHIMERE model is a one-way chemistry-transport model: the simulations for the parent domain

influence the simulations for the nested domain, however, the inverse is not applied; therefore any concentrations observed in the nested domain will not change the concentrations seen in the parent domain.

Examining figures corresponding to OA concentrations (Figures 12-a and 12-b), at the surface level there is a region of high concentration of OA in the Gulf of Genoa (between Genoa and Corsica) reaching nearly 4 µg.m$^{-3}$. The high concentration in this area is less pronounced in the BL (Figure 12-b). The concentration of biogenic OAs



(Figures 12-c and 12-d) is high over the basin as well as over Europe, and less important over North Africa. The percentage of contribution of this type of aerosol to the overall OA concentration is shown in figure 13-a and 13-b. As expected from looking at the total concentration of biogenic OAs, their contribution stays on average around 70% over the basin, but is lower in the Gulf of Genoa (about 60% at surface, nearly 70% at altitude). In this area

the secondary anthropogenic organic aerosols show higher contributions compared to their contribution to the rest of the domain (around 14% instead of 12%), also, the contribution of primary organic aerosols (and its subsequent oxidations) is also quite high in this region (around 20%). The areas between the Corsica and Marseilles and also the northern coasts of Africa are main shipping routes in the western Mediterranean basin, with high amounts of shipping related emissions. They affect primary organic aerosol POA and its semi-volatile oxidation products as

shown in figure figures 12-g and 12-h, with concentrations as high as 1.5 µg.m$^{-3}$ around Corsica and 1 µg.m$^{-3}$ near the African coasts. This corresponds to a contribution of around 30% over Corsica and 22% over the coasts of Africa (Figures 13-e and 13-f). These values may actually be upper limits as shipping primary SVOC and IVOC emissions were treated as those from other activity sectors (see section 2.2.3), and as recent chamber study data suggest lower SOA yields from shipping emissions (Pieber et al., 2016). The influence of shipping emissions is

not visible in the BL, because vertical mixing within the marine boundary layer is weak, thus in the 300 to 400m layer the contribution of biogenic OA becomes dominant for the whole basin. Still, in the Gulf of Genoa, some effects of anthropogenic influences are visible in the BL, which could be linked to emissions originating from industrial sites and the harbor of Genoa area, which are apparently better mixed vertically over the continental convective boundary layer (in the coastal region with strong orography). These anthropogenic emissions are visible

in figures corresponding to anthropogenic OA, formed from anthropogenic VOC's and especially aromatic compounds (Figures 12-e and 12-f for absolute concentrations and figures 13-c and 13-d for relative percentages). While the average concentration of anthropogenic OAs stay relatively low over the western Mediterranean basin (an average of 0.30 µg.m$^{-3}$ near the surface), they become more pronounced around the Gulf of Genoa and the eastern part of the domain both near the surface and in the BL. A contribution of about 12-15% both near the

surface and in the BL is seen for this component above the Gulf of Genoa and over the highly industrialized and densely populated Po Valley.

**7 Conclusion/discussion**

Three schemes for the simulation of organic aerosol were implemented and tested along with the standard scheme in the CHIMERE chemistry-transport model. The simulations from each of the four schemes were compared to

detailed experimental data obtained from two different stations in the western Mediterranean area during the ChArMEx campaign in summer 2013. The simulations were performed on 4 nested domains with increasing resolutions, the largest one covering Europe and northern Africa with a 30-km horizontal resolution, to the smallest one focused on the Cap Corse area with a 1km horizontal resolution.

For the comparisons of OA simulated with different schemes to observations, we explored three different aspects:

mass concentration, distribution with respect to volatility and oxidative state of OA classes derived by PMF, and $^{14}$C measurements discriminating fossil or non-fossil origin. Results show that the modified VBS scheme (i.e. including the fragmentation and formation of non-volatile organic aerosols), better corresponds to the observations at both sites, and this for all three aspects. The modified VBS scheme succeeds at simulating the average concentration of OA for the one-month campaign period with low bias (about -20% at both sites), even if the





hourly variability is not perfectly displayed (as for other aerosol components). Comparisons for OA precursors (isoprene and terpenes) and isoprene oxidation products (sum of methyl vinyl ketone and methacroleine) were performed and showed significant differences, which do however not necessarily affect the model ability to form BSOA, because the BVOC measurements are representative for a local scale, while BSOA formation occurs on a larger regional scale.

Furthermore, the fossil/non-fossil distribution of OA was explored in different schemes. The modified VBS scheme corresponds better to available data in this regard as well. It is also the only scheme among the four tested that represents the distribution of the different PMF factors in a satisfying way attributing the major OA fraction to LV-OOA (but slightly overestimating this part). The differences between the sites, especially the more local anthropogenic character of the Es Pinar site (larger 14C fossil fuel origin, larger SV-OOA fraction, higher NOx concentration which is a good tracer for anthropogenic pollution) and the lower OA concentration at this site are qualitatively simulated well with this scheme. While the standard VBS scheme without BSOA aging is as close to mass and origin comparisons as the modified VBS scheme, by construction, it does not include the formation of LV-OOA, resulting in an important overestimation of the SV-OOA at both sites.

A closer look at OA sources over the western Mediterranean basin simulated with the modified VBS scheme, selected because of its good results (at least for the summer of 2013 period and for this given region), shows that the OA with biogenic origins is dominant in the whole basin. In areas between Corsica and Marseilles, the golf of Genoa and also the northern coast of Africa, the contribution of biogenic organic aerosols is less than for other parts. This fact points to the influence of shipping emissions for the areas between Marseilles and Corsica and also the northern coast of Africa, which can be seen in the contribution of POA and their oxidation products to the formation of OA over the basin, even if this part may be overestimated in the current simulations. For the gulf of Genoa, slightly higher contribution of anthropogenic organic aerosols was observed compared to other parts of the domain. However, at a higher altitude, the contribution of the biogenic sources becomes dominant in the whole basin, with a significant drop in the contribution of POA over the basin and leaving only a small trace of anthropogenic contribution in the gulf of Genoa. This contribution is attributed to ASOA rather than POA for the industrial area in the northern Italy, which is a persistent source of ASOA both near the surface and at higher altitudes.

It would be useful to compare the precursor components for the formation of OA such as isoprene and monoterpenes to multiple stations rather than only two, since the measurements for these species tend to have a more local profile rather than a regional one, therefore multiple stations spread in the domain would give the needed regional aspect of the comparisons. Longer periods of simulations with comparisons to observations are also necessary, since processes leading to the formation of OA can change in other seasons and especially in winter when the biogenic contribution is much lower. Since airborne measurements for OA and biogenic gas-phase precursors were also performed during the summer of 2014 in the SAFMED+ campaign over different forested areas, it would be useful to continue the simulations for this year to compare the results to in-situ measurements and airborne measurements at the same time.

**Acknowledgements.** This research has received funding from the French National Research Agency (ANR) projects SAF-MED (grant ANR-15 12-BS06-0013). This work is part of the ChArMEx project supported by ADEME, CEA, CNRS-INSU and Météo-France through the multidisciplinary programme MISTRALS



(Mediterranean Integrated Studies aT Regional And Local Scales). The station at Ersa was partly supported by the CORSiCA project funded by the Collectivité Territoriale de Corse through the Fonds Européen de Développement Régional of the European Operational Program 2007-2013 and the Contrat de Plan Etat-Région. The EEA is acknowledged for air quality data for several stations in Europe which was used for observation-simulation

comparisons. The NCEP is acknowledged for the meteorological input data used in the WRF meteorological model. E-OBS datasets is acknowledged. The thesis work of Arineh Cholakian is supported by ADEME, INERIS (with the support of the French Ministry in charge of Ecology), and via the ANR SAF-MED project. The ChArMEx team is acknowledged for his great help in organizing the measurement campaign at Ersa, as well as the MISTRALS management and accounting team. Jorge Pey is currently granted with a Ramón y Cajal research

contract (RYC-2013-14159) from the Spanish Ministry of Economy, Industry and Competitiveness.

This work was performed using HPC
resources from GENCI-CCRT
(Grant 2017- t2015017232).

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





**Tables**

| Domain name | Cell numbers | Resolution |
|---|---|---|
| D30 | 71×64 | 30km×30km |
| D10 | 110×110 | 10km×10km |
| D3 | 110×110 | 3km×3km |
| D1 | 80×80 | 1km×1km |

**Table 1.** List of domains with their respective horizontal resolutions, the domains are shown in figure1.



1: Ersa, France (42°58'04.1", 9°22'49.1")

**Gases:** NO$_x$ (CRANOX), VOCs (PTR-ToF-MS-KORE)

**Aerosols:** PM$_{10}$ total mass (TEOM-FDMS), On-line (non refractive) PM$_1$
Chemistry : OA, SO$_4^{2-}$, NH$_4^+$, NO$_3^-$ (ACSM), $^{14}$C (PM$_1$, daily filters),
BC (PM$_{2.5}$, MAAP)

**Meteorology:** temperature, wind speed, wind direction, relative humidity

2: Es Pinar, Spain (39°53'04.6", 3°11'40.9")

**Gases:** NO$_x$ (CRANOX), VOCs (PTR-ToF-MS, Ionikon)

**Aerosols:** PM$_{10}$ total mass (BETA corrected by factor obtained after
comparison with gravimetric), PM$_1$ on-line Chemistry: OA, SO$_4^{2-}$,
NH$_3^+$, NO$_3^-$ (HR-ToF-AMS), $^{14}$C (PM$_1$, daily filters),
BC (PM$_{2.5}$, Aethalometer AE33)

**Meteorology:** temperature, wind speed, wind direction, relative humidity

3 : Palma, Spain (39° 36' 20.88", 2° 42' 23.7594"), 4 : Nîmes, France (43° 51' 21.6", 4°
24' 21.5994"), 5 : Ajaccio, France (41° 55' 5.3256", 8° 47' 38.0538")

Radio-soundings : T, RH, wind speed, wind direction; at 00 and 12 UTC each day

**Table 2.** Gas/aerosol and measurements used for this study.





| Pollutant | ORE (%) | Pollutant | ORE (%) | Pollutant | ORE (%) | Parameter | ORE (%) |
|-----------|---------|-----------|---------|-----------|---------|-----------|---------|
| $O_3$ | 4 | $C_5H_8$ | 85 | Mono-terpenes | 59 | Temperature | 0.5 |
| OA | 10 | BC | 26 | $SO_2$ | 62 | Relative humidity | 18 |
| $SO_4^{2-}$ | 15 | $NO_x$ | 75 | Aromatic species | 49 | | |
| $PM_{10}$ | 9 | CO | 2 | MACR+MVK | 60 | | |

**Table 3.** Calculated relative orographic representativeness error (ORE) for a list of species and meteorological parameters. MACR+MVK presents the sum of methyl vinyl ketone and methacrolein.



| | | Ersa | | | | Es Pinar | | |
|---|---|---|---|---|---|---|---|---|
| | ORE | R | RMSE | Bias | Mean_obs | R | RMSE | Bias | Mean_obs |
| | % | ppb | | | | | | | |
| NO$_x$ | 75 | 0.37 | 0.68 | 0.16 | 0.62 | 0.12 | 2.76 | -1.51 | 3.53 |
| C$_5$H$_8$ | 85 | 0.76 | 0.44 | 0.24 | 0.19 | 0.69 | 0.10 | -0.04 | 0.17 |
| MACR+MVK | 60 | 0.62 | 0.22 | 0.09 | 0.09 | 0.41 | 0.09 | 0.005 | 0.10 |
| Mono terpenes | 59 | 0.35 | 0.69 | -0.38 | 0.52 | 0.14 | 0.11 | -0.06 | 0.09 |

**Table 4.** Statistical data for time series shown in figure 6, Mean_obs shows the average of observations





| | ORE | Ersa | | | | Es Pinar | | | |
| | | R | RMSE | Bias | Mean_obs | R | RMSE | Bias | Mean_obs |
| | % | $\mu g.m^{-3}$ | | | | | | | |
| BC | 26 | 0.36 | 0.16 | 0.02 | 0.39 | 0.47 | 0.22 | -0.13 | 0.39 |
| $SO_4^{2-}$ | 15 | 0.42 | 1.72 | 1.21 | 1.90 | 0.52 | 1.93 | 0.91 | 2.70 |

**Table 5.** Statistical data for time series shown in figure 6, Mean_obs shows the average of observations.



| | | CHIMERE standard | | Standard VBS ba | | Standard VBS nba | | Modified VBS | |
|---|---|---|---|---|---|---|---|---|---|
| | | Ersa | Es Pinar | Ersa | Es Pinar | Ersa | Es Pinar | Ersa | Es Pinar |
| R | | 0.46 | 0.16 | 0.50 | 0.11 | 0.45 | 0.22 | 0.50 | 0.29 |
| RMSE | | 3.96 | 2.65 | 9.63 | 7.29 | 2.17 | 1.89 | 1.59 | 1.49 |
| Bias | $\mu g.m^{-3}$ | 3.39 | 2.49 | 9.05 | 7.17 | 1.26 | 1.25 | -0.68 | -0.65 |
| Mean_sim | | 7.10 | 5.36 | 12.76 | 10.17 | 4.97 | 4.13 | 3.02 | 2.22 |
| Mean_obs | | 3.71 | 2.88 | 3.71 | 2.88 | 3.71 | 2.88 | 3.71 | 2.88 |

**Table 6.** Statistical data for hourly time series shown in figure 7. R is the correlation coefficient between model and measurements, RMSE the root mean square error, mean_sim and mean_obs show the average of simulations and observations respectively.



|  | Repartition in model |
|---|---|
| Fossil | ASOA, BC (96.4%), POA (87.7%) |
| Non fossil | BSOA, BC (3.6%), POA (12.3%) |

**Table 7**. Distribution of model species into fossil/non-fossil fractions (Sasser et al., 2012)



**Figures**

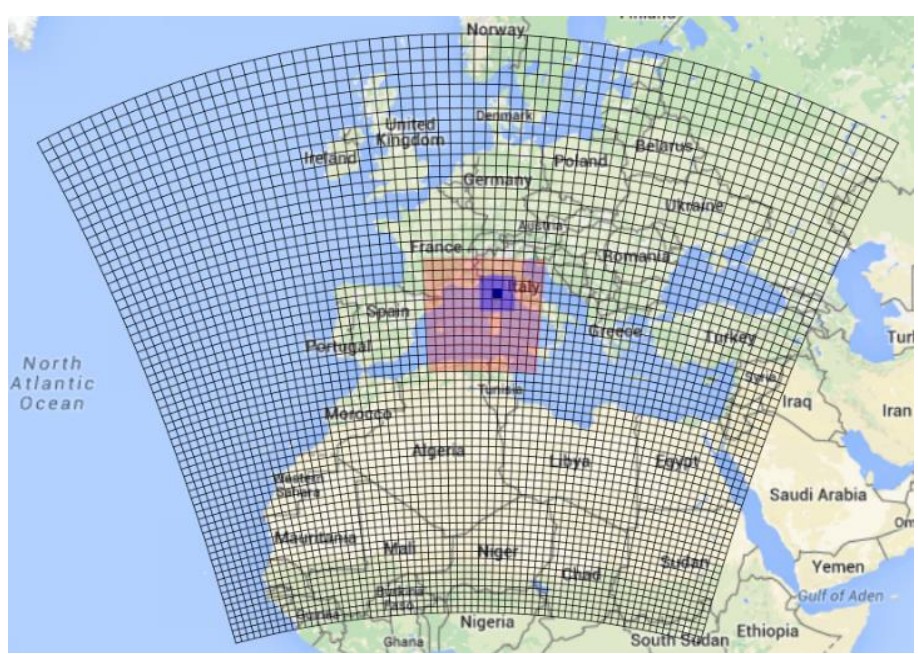

**Figure 1.** Domains used in simulations, in order of size: D30, D10, D3, D1. The resolution for each domain is given in table 1.





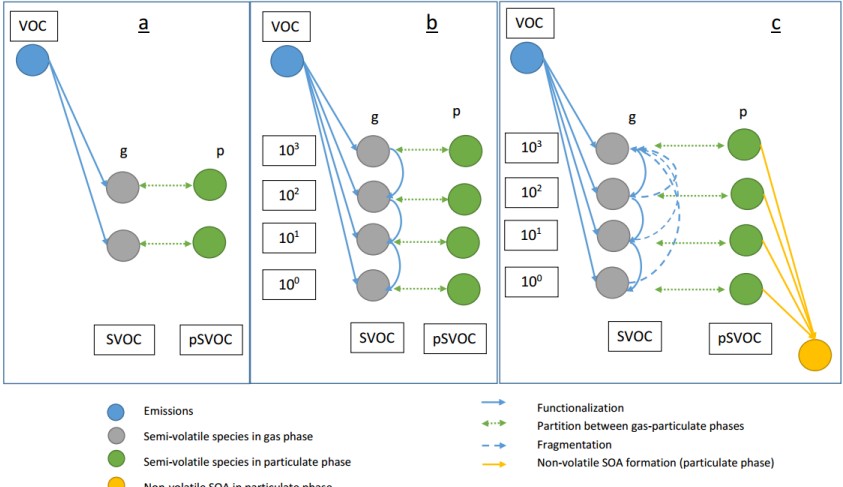

**Figure 2.** OA simulation schemes - a: CHIMERE standard scheme (Bessagnet et al., 2008): from a parent VOC, different semi-volatile VOC (SVOC) compounds (only one represented) are formed in a single step by oxidation; they are in equilibrium between gas and aerosol phase (pSVOC); b: VBS standard scheme (Robinson et al., 2007): from a parent VOC, SVOC with regularly spaced volatility ranges are formed and are in equilibrium with the aerosol phase. Aging of SVOC by functionalization is included by passing species to classes with lower volatility; c: modified VBS scheme (Shrivastava et al., 2015): here SVOC aging also includes fragmentation, leading to transfer of species to classes with higher volatility. In addition, semi-volatile aerosol can be irreversibly transformed into non-volatile one (yellow-filled circle). For each bin saturation concentration is shown in $\mu g.m^{-3}$.



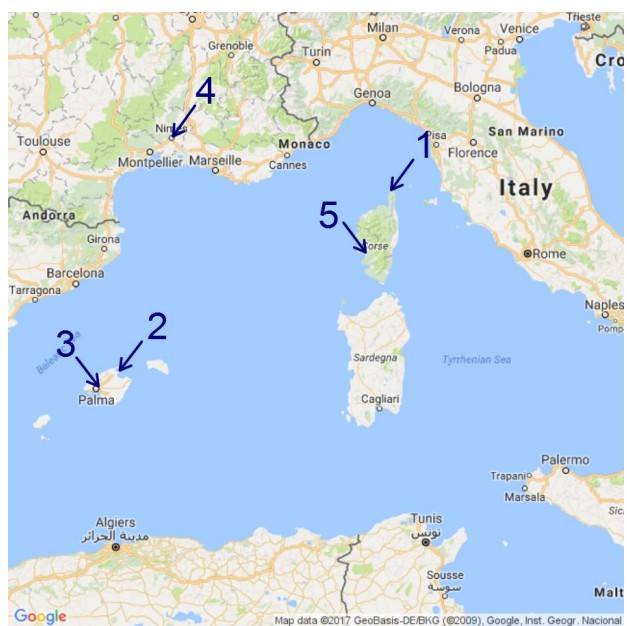

**Figure 3.** Experimental data sites, more details in table 2. 1: Ersa, France, 2: Cap Es Pinar, Spain, 3: Palma, Spain,
4: Nîmes, France, 5: Ajaccio, France


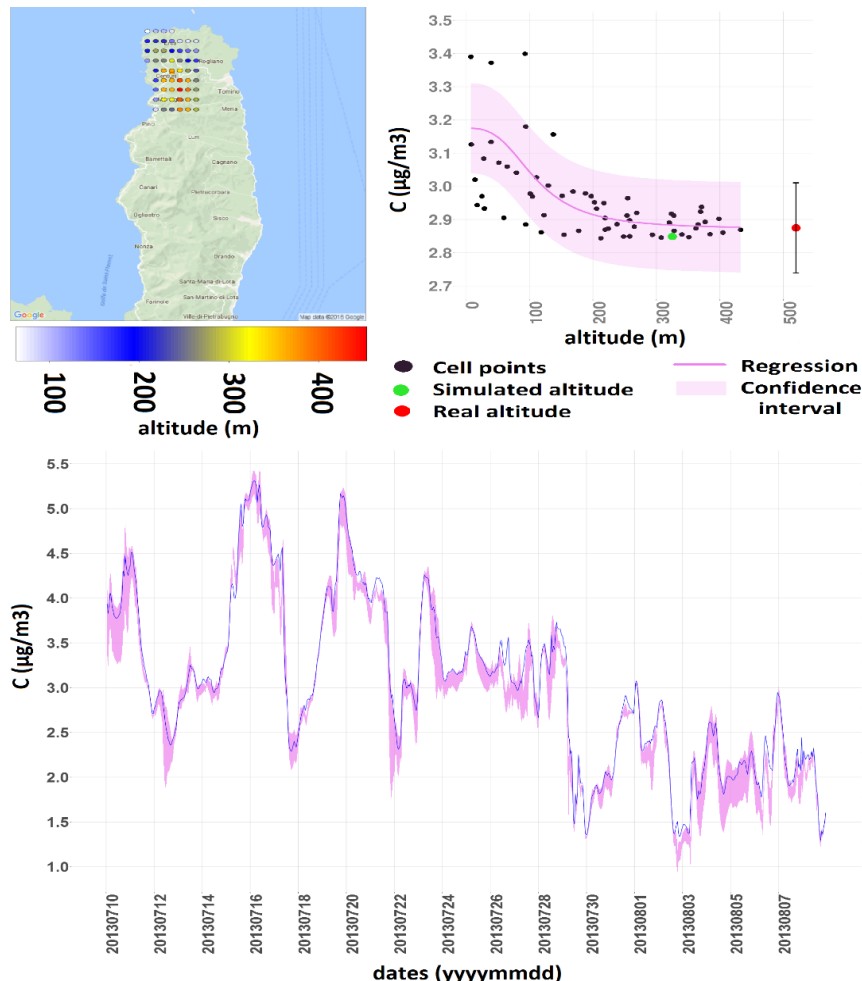

**Figure 4.** Orographic representativeness error- a: neighbouring cells used in the orographic representativeness test; b: an example of non-linear regressions performed on one time-step for OA (1 point corresponds to 1 grid cell); c: results for all hourly simulation times for OA. In b and c, the purple ribbon shows the confidence interval of the regression results. In c, the blue line shows the simulations at the nominal Ersa site grid cell.



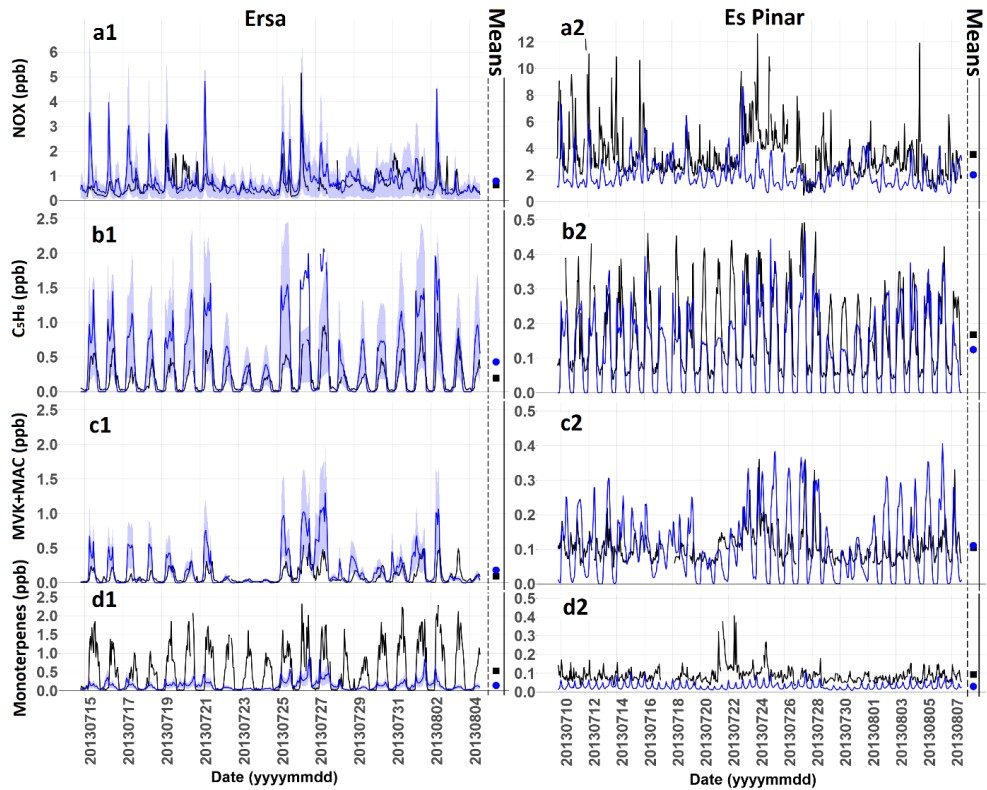

**Figure 5.** Time series showing the comparison of simulated (in blue) and measured (in black) gaseous species during the ChArMEx/SAFMED campaign period. a: nitrogen oxides; b: isoprene; c: MACR+MVK, d: monoterpenes. Statistical data for these comparisons are given in table 4. Beside each time table two points are shown presenting the average for simulations (in blue circle) and observations (in black square). Ribbon around Ersa simulations presents orographic representativeness error.





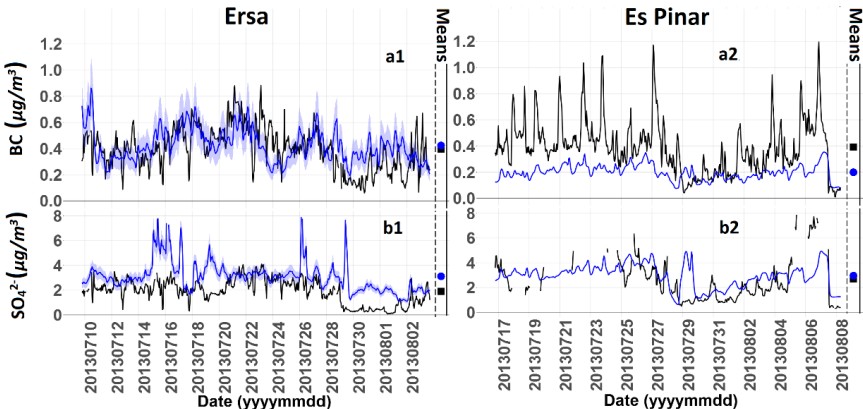

**Figure 6**. As for figure 5, but for particulate matter. a: BC; b: sulfate particles. Statistical data for these comparisons is given in table 6.





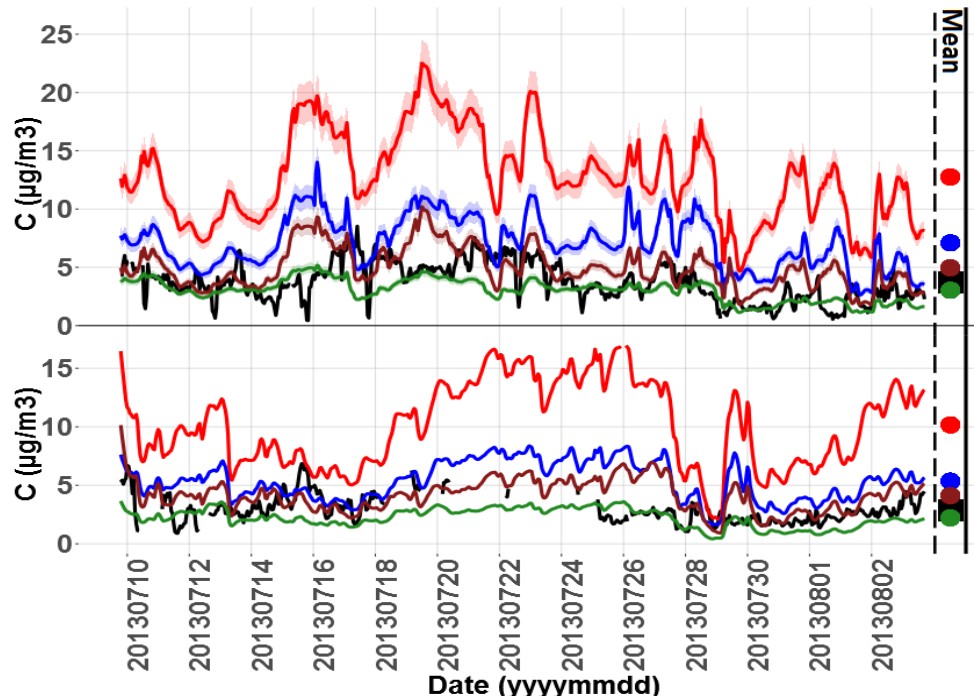

**Figure 7.** Compared time series of total $PM_1$ OA concentration at Ersa (top) and Es Pinar (bottom) – black:observations; green:modified VBS; brown:standard VBS without BSOA aging; blue:CHIMERE standard; red:standrad VBS with BSOA aging. For each site the average of simulations is shown with circles in corresponding colors and the observations with black squares. Shaded area for Ersa presents the orographic representativeness error. Statistical data is shown in table 6.





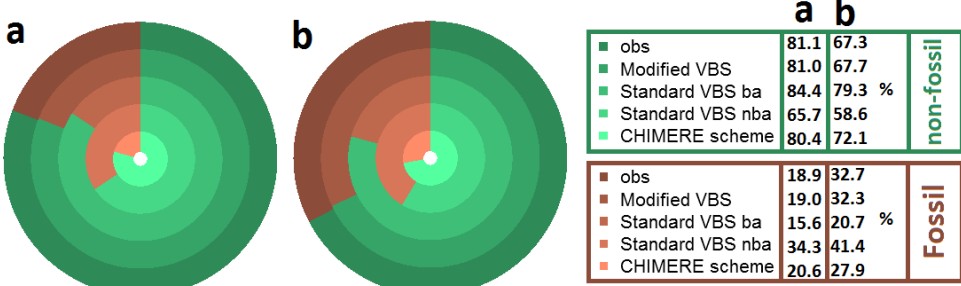

**Figure 8.** Comparison of simulations to 14C measurements - a: comparison of filters to four schemes for Ersa; b: comparison of filters to four schemes for Es Pinar. For both figures from outer ring to inner ring: observations, modified VBS, standard VBS with BSOA aging, standard VBS without BSOA aging and CHIMERE standard scheme.



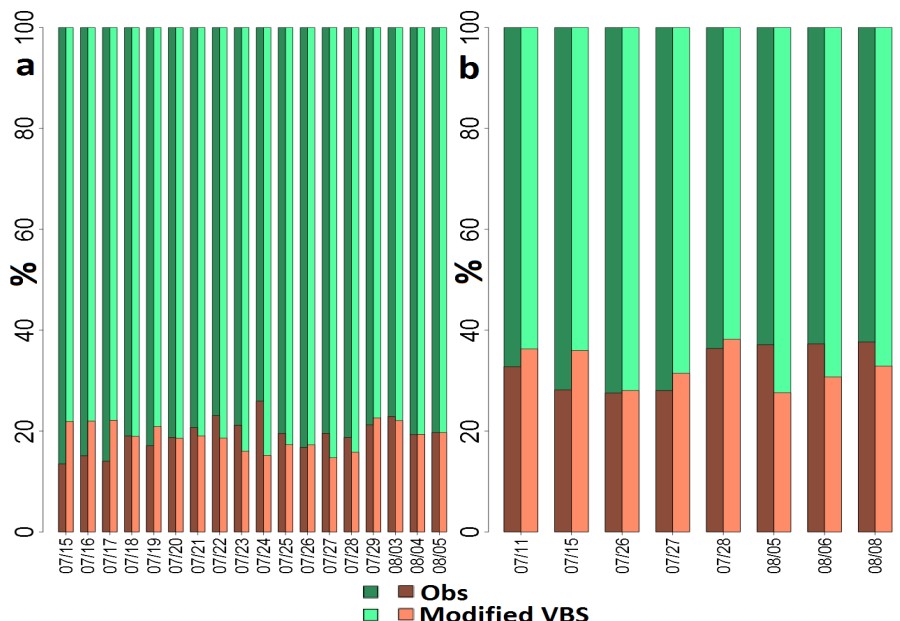

**Figure 9.** Comparison of simulations to $^{14}$C measurements - a: Ersa individual filters; b:Es Pinar individual filters. For each filter, measurements are on left and modified VBS simulations are on right. Dark/light brown shows fossil and dark/light green shows non-fossil sources.



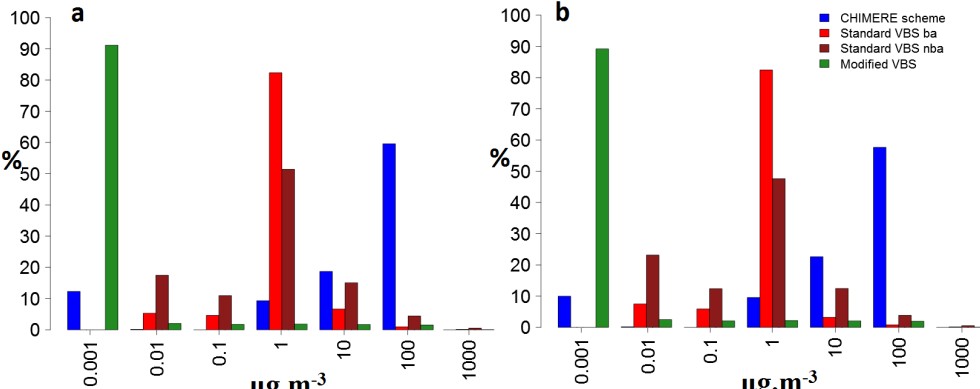

**Figure 10.** Distribution of simulated OA in volatility bins - a: Ersa volatility bins; b:Es Pinar volatility bins.





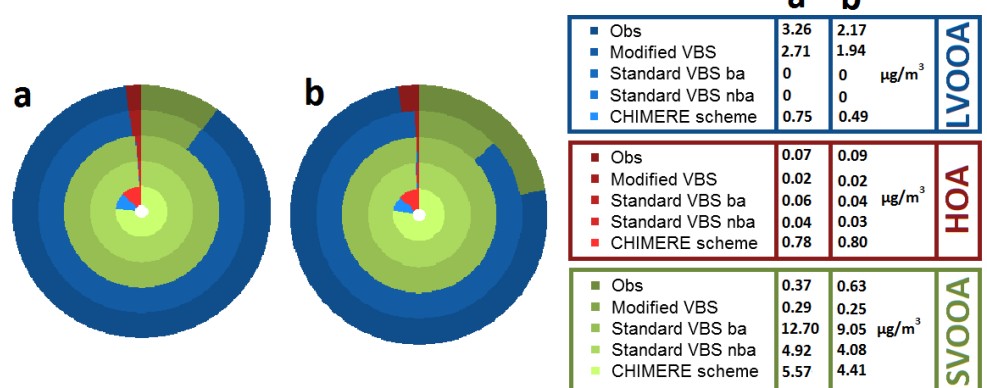

**Figure 11.** Comparison to PMF results - a: comparison of observed PMF factors to that derived from four schemes for Ersa; b: for Es Pinar. For both figures from outer ring to inner ring: observations, modified VBS, standard VBS with BSOA aging, standard VBS without BSOA aging and CHIMERE standard scheme.




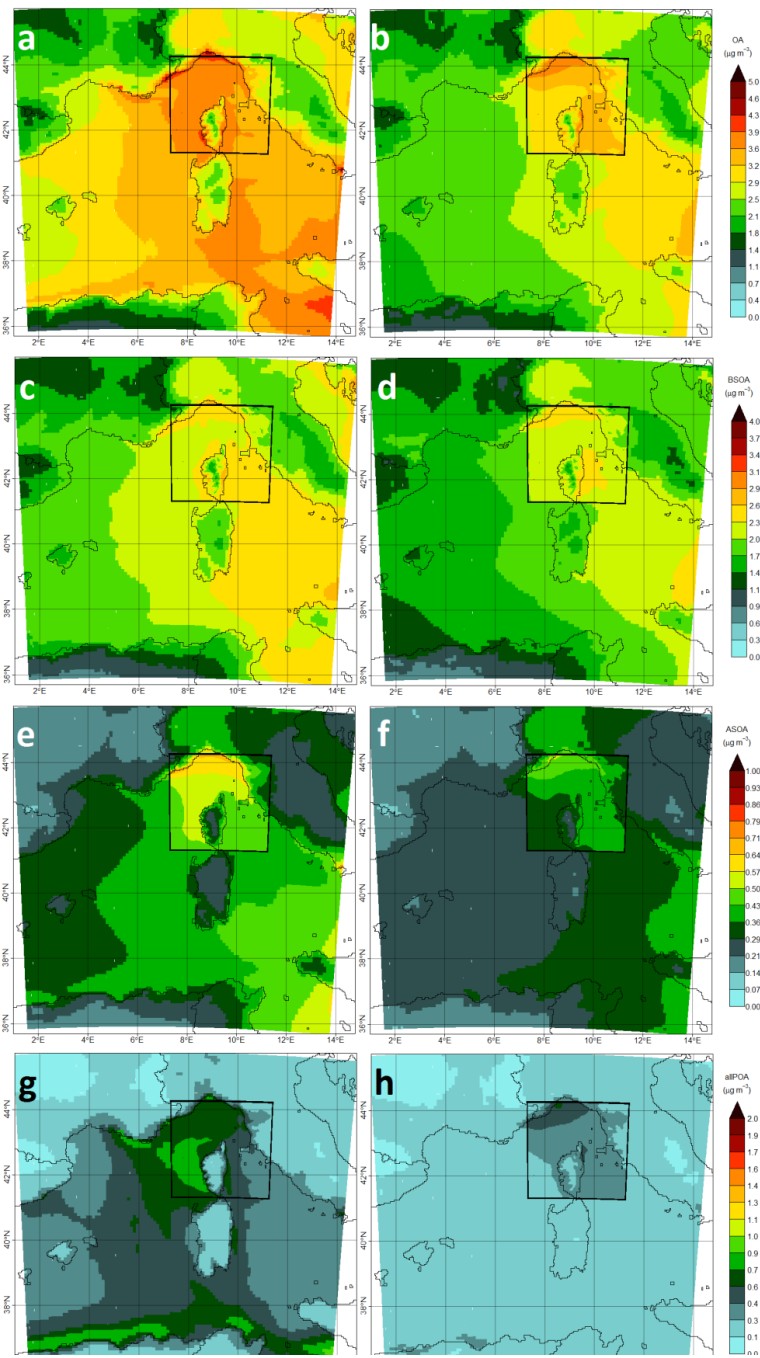

**Figure 12.** Organic aerosol concentrations over the western Mediterranean basin. Simulations were performed with the modified VBS scheme. Left column is near the surface, and right column for an altitude between 300-450m asl. a-b: total OA concentration($\mu$g.m$^{-3}$); c-d: BSOA concentration ($\mu$g.m$^{-3}$); e-f: total ASOA concentration($\mu$g.m$^{-3}$); g-h: sum of POA and its subsequent oxidations products($\mu$g.m$^{-3}$). Results are from the D3 simulation (3-km horizontal resolution) within the framed area, from the D10 simulation outside (10-km resolution).



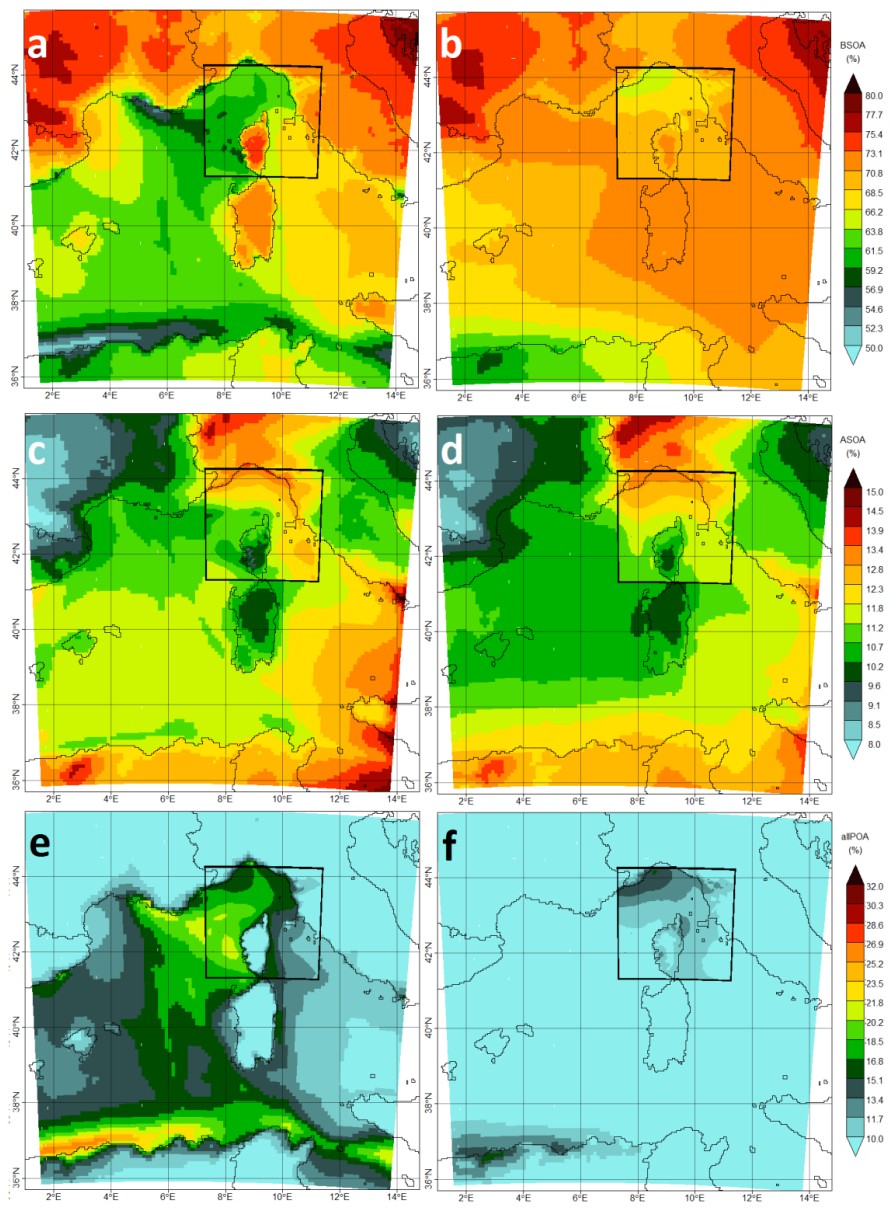

**Figure 13.** As figure 12 but for relative contributions of BSOA/ASOA/POA to total OA (%) a-b: BSOA; c-d: ASOA; e-f:POA and its subsequent oxidation products. a-b: biogenic secondary organic aerosol (BSOA); c-d: anthropogenic secondary organic aerosol (ASOA); e-f: POA and its subsequent oxidation products.