# Peer review of "Simulation of fine organic aerosols in the western Mediterranean area during the ChArMEx 2013 summer campaign"

_Atmospheric Chemistry and Physics, 2017_

## Referee Comment (RC1) · Anonymous Referee #1 · 13 Nov 2017

This manuscript present a study concerning Secondary Organic Aerosols (SOA) over the western Mediterranean basin using the Chemistry Transport Model (CTM) CHIMERE. This study is done in the framework of the ChArMEx project. The aim of the study is to compare four different ways of modelling SOA in the CHIMERE modele using SOA measurements made during ChArMEx, especially the SAFMED campaign. After an introduction, the manuscript presents the CHIMERE model and the four SOA schemes used in this study. Section 3 presents the observation used in this study for the validation of the model simulations done in section 4 and 5. The selected SOA scheme, being the modified VBS is then used to simulate the organic aerosol over the western part of the Mediterranean basin in order to examine the composition of the

organic aerosols.

This manuscript is well presented and give a extended validation of the organic aerosols over the two main ground stations used during the SAFMED campaign. The authors deal with the issue of the orography that appears here over the Ersa station. The solution provided is original and very interressant. The sentences are sometimes too long. They may be hard to understand sometimes.

I would then recommend minor revisions before publication.

**General comments:**

- Page 6, line 2: I don't understand the aim of the sentence "Four volatility bins are added for anthropogenic and biogenic SOA ranging from 1 to 1000µg.m-3". You say on page 5, that you have 9 volatility bins between 0.01 and 106, corresponding to the figure SI2. On the Figure 2, you represent the 4 volatility bins between 1 and 1000. Which bins do you use in the model? Please make the manuscript clearer.
- 2. Page 10, line 34: I don't understand what you mean by "for regimes".
- 3. Page 12, line 18-20: Could you explain a little more the cloud effect affecting aerosols concentrations.
- 4. Page 18, line 10-12: The concentrations of allPOA around Corsica does not reach  $1.5 \mu g.m^{-3}$  in figure 12.g. Same for figure 13.e and the 30% of allPOA to the OA. Could you correct the text.

**Figure related comments:**

1. Figure 4: You mention a, b and c, but they are not reported on the figure.
2. Figure 5.b1: The blue ribbon around the date 20130726 seems weird. As I understood your method, it should be placed around the simulation results (blue line), but on these part of the graph it seems to be around the black line corresponding to the observation. Could you explain why?

Technical comments (when a letter or a word is missing, it is in **bold** in the comment):

- 1. Page 5, line 26: Please add "SI1" after "in the supplementary information".
- 2. Page 6, line 33: as in Shrivastava et al. 2013 -> as in Shrivastava et al. (2013)
- 3. Page 6, line 36: life time -> lifetime
- 4. Page 10, line 14: Please add "Orographic Representativeness Error" for the clarity of the text.
- 5. Page 12, line 30: You say the correlation is 0.26, but in the table it is 0.36 for Ersa and 0.47 for Es Pinar. Could you correct, or explain?
- 6. Page 14, line 22: Could you precise what US stands for?

**ACPD**

---

## Referee Comment (RC2) · Anonymous Referee #2 · 21 Nov 2017

This paper describes efforts to simulate atmospheric meteorology, gas-phase species, and particle-phase species during the ChArMEx 2013 campaign. Surface meteorology data are derived from a site on Corsica and a site on Mallorca. More elevated meteorology data are derived from soundings on the continent. Gas-phase and particle-phase data are derived from the surface sites on Corsica and Mallorca. The orographic nature of the Corsica site caused an additional complexity within the model. The modeling effort uses four different techniques to simulate organic aerosol loadings, and that which best matches observations (at the two surface sites) is used to apportion the organic aerosol to primary material (assumed predominantly to be anthropogenic) and biogenic/anthropogenic secondary material. At these locations, biogenic secondary or-

ganic aerosol seems to dominate, at least during the period of observation/simulation. This is not a surprising result based on previous simulations in Europe.

While I recognize the importance of being able to simulate secondary organic aerosol, this paper seems more like a method development paper in that it extensively compares the results from the four SOA techniques. However, these types of evaluations were done during development of those models. As a result of these efforts, the authors recommend the updated VBS approach which takes into account fragmentation, as well as formation of non-volatile material. This is not surprising given the documented issues with the more simple CHIMERE standard approach and the VBS with the biogenic aging. In addition, the model does not include any cloud processing, despite the fact that the observations indicate that such a phenomenon is likely to occur.

In addition, the authors do not give appropriate credit of the work of Chrit et al. (2017) who simulated this same data, using a different approach. The authors indicate that the concentrations and properties are well simulated – so this calls into question the need for the current paper. What does this new study tell us that the work of Chrit et al. (2017) did not?

As a result of these facts, I do not find this manuscript very novel, and I am unable to recommend publication. However, if the authors wish to completely refocus the manuscript, they must address the issues below, in addition to the novelty issues. The authors should consider also simulation of other time periods (as suggested in Page 19, Line 28, the last paragraph).

The paper also could use some editing for writing. There are several instances where verbs are missing and where coordinating conjunctions are missing (particularly when the word therefore is used). There are also some word choice issues (sanitary, transportation), and it should be noted that the word 'data' is plural so that verb agreement needs to be corrected. There are also instances where paragraphs only have one sentence.

Other specific comments Section 2 Page 4, Line 8. The wording here makes it seem like Europe is a country.

Page 4, Line 13. I did not find Table 1 particularly useful. It could be incorporated into the Figure 1 caption.

Page 4, Line 33. What is a shipping snap sector?

Page 5, Line 10. Has MELCHIOR been updated since its inception in 2003?

Page 5, Line 13. What is the distribution of particle sizes in the model? That is, the authors provide the range but not any information about how the bins are spaced.

Page 5, Line 24. Pun and Seigneur were not the source of the experimental data. This citation should be to the original manuscripts.

Page 6, Line 27. Please provide more detail on how this ratio of NO reaction rates is used to determine the low versus high NOx yields. What values of this ratio correspond to low versus high NOx? What are the bounds of this ratio?

Section 3 Page 7, Line 19. Was only total NOx measured and compared? Were there any issues due to conversion of other NOy species into NO?

Page 8, Lines 2 and 5. I do not understand why a constant value of HOA was assumed for each PMF. This needs to be clarified as model results are later compared to these values.

Section 4 Page 9, Line 6. The wording here should be changed to indicate that it is the simulated height of the cell in which the site was located was underpredicted – not that the simulated height of the site was underpredicted (since the authors are not actually simulating the height of the site).

Page 9, Line 27. Higher than a fixed value? What value? What was its basis?

Page 9, Line 39. The authors state that secondary species appear less influenced by

the orographic uncertainty. However, the impact on MVK + MACR is as large as that for primary species. The authors need to be careful with their word choice.

Page 10, Line 35. I do not believe that this last sentence is necessary.

Page 11, Line 7. The caption from Table 4 needs to refer to Figure 5.

Page 11, Line 32. Does this requirement of a regional look at the aerosol call into question the use of two sites?

Page 12, Line 19. Here is evidence for cloud processing (again, it is presented for BC later). However, no mention of cloud processing in the aerosol modules is mentioned. Why not link the updated VBS with a cloud processing module? That would certainly enhance the novelty of the work.

Page 13, Line 3. I do not find this summary paragraph necessary.

Page 13, Line 27. Figure 7 caption has an error in the word standard.

Page 13, Lines 30 and 31. Please provide standard deviations on the averages.

Page 14, Line 12. What is meant by 'respected'?

Page 14, Line 34. I do not see much use for Table 7. This text can be included in the main body of the manuscript.

Page 15 and beyond, The colors in Figures 8, 9, and 11 need to be differentiated to a greater extent if they are included in any future submission.

Page 17, Line 15. I find this summary paragraph unnecessary.

---

## Short Comment (SC1) · 12 Dec 2017

Dear editor, dear referee,

In this reply, we would like to respond to some of the fundamental issues raised by one referee (the other being positive about the paper requesting only minor modifications) on novelty and also similarities to the paper by Chrit et al.. We thank the referee for initiating this discussion, but we hope and think being able to show the originality of our work. We will respond to the detailed remarks of both referees in a later stage. The referee's statements are numbered and are in quotes.

[Figure]

1 – "In addition, the authors do not give appropriate credit of the work of Chrit et al. (2017) who simulated this same data, using a different approach. The authors indicate that the concentrations and properties are well simulated – so this calls into question the need for the current paper. What does this new study tell us that the work of Chrit et al. (2017) did not?"

Although the two papers were based on exploiting results from the same measurements campaign, there are fundamental differences between them. Apart from obvious differences like different models with different inputs, the approach taken in the two articles regarding the simulation of organic aerosols is quite different. In our article, we use the VBS scheme with modifications that take into account formation of non-volatile SOA and fragmentation processes. In Chrit et al. (2017), they use a surrogate based approach where SOA are divided into three types, hydrophilic, hydrophobic or both (Couvidat et Sartelet 2015) with modifications added to take into account ELVOCs. In our mind, it is important to compare all these different schemes to detailed measurement data and not limit this comparison to one particular scheme.

Our paper goes significantly beyond the Chrit et al. (2017) paper. In our work, we use in addition to data from Cap Corsica measurements from a second site at Mallorca for the same period. The two sites have different characteristics as it is seen in the results of 14C measurements and PMF results, the one at Mallorca being more strongly influenced by anthropogenic emissions. Since the goal of the article was to evaluate the performance of different SOA simulation schemes in the western Mediterranean area, it is more representative to take into account campaign data from the two sites. Measurements used for Mallorca station have not been compared to simulations in other articles (neither the OA concentration, 14C measurements, nor the PMF results), so this point is clearly novel. In addition, our work provides, in addition to that of OA, a detailed comparison of meteorological inputs, SOA precursors (isoprene and mono-terpenes) and intermediate compounds (MVK+MAC) for both sites and thus goes beyond Chrit et al. (2017).

[Figure]

For the Cap Corsica station, because of its unique geographical characteristics not well represented by CTMs, an orographical representativeness error study is performed in our work, which is actually sited in Chris et al. (2017) to indicate an error for organic aerosol (Chrit et al. 2017, Page 7, section 4, sub-section 2). The novelty of this approach is discussed in more detail below.

A source apportionment study for the western Mediterranean basin at two different altitudes (surface and 300-450m altitutes) is also included in our study, which was not included in Chrit et al. (2017). This study shows the impact of biogenic and anthropogenic sources for the formation of OA not only on the surface, but also for a higher altitude.

As a side note, the two articles were written at almost the same time period. While Chrit et al. (2017) was accepted for publication on 23 October 2017, this article was submitted for discussion on 26 august 2017.

2 – "As a result of these facts, I do not find this manuscript very novel."

Here, we wish to show that our work is indeed novel and original in several important aspects.

The Cap Corsica site has unique geographical characteristics which raises issues when comparing simulations to measurements. Indeed, it is located on the northerly edge of a crest and surrounded from the west, north and east by slopes falling rapidly to the sea. Its altitude is at 530 m, while in CHIMERE simulations, even at a 1 km horizontal resolution, it is represented at 360 m height. This height difference might induce differences between simulated and observed concentrations, which needs to be assessed, especially in a marine environment with low boundary layer heights. Therefore, a novel approach was developed in this work to calculate an orographic representativeness error, based on comparing simulated concentrations at horizontally very close grid cells located at different altitudes (from 100 to 500 m). The method allows attributing representativeness error bars to different species, large for primary

compounds especially emitted in the marine boundary layer like NOx and BC, smaller for (mainly) secondary compounds as O3 and OA, which are compared to simulation observation differences. As far as to our knowledge, this approach was not taken anywhere else for the estimation of errors produced by orographic representativeness of a site. This method can be applied to other sites with the same characteristics (high altitude remote sites with strong change of altitude compared to nearby terrain). The results obtained from this approach are also cited in Chrit et al 2017 (Page 7, section 4, sub-section 2). We would argue that this development, while not the primary goal of the paper, is clearly novel.

For the eastern Mediterranean area, an abundance of literature is available both for chemical composition (Bardouki et al., 2003 ; Koçak et al., 2007b ; Koulouri et al., 2008 ; Hildebrandt et al., 2010) and also for OA comparisons to simulations (Fountoukis et al., 2014). However, for the western part of the basin, much less studies exist for the chemical composition of the area (for example Querol et al., 2009 ; Pey et al., 2013), but these data have not been used to the best of our knowledge specifically for CTM model evaluation. Thus, detailed model -observation comparisons of different aspects of OA (total concentrations, oxidation states, modern/fossil fractions from 14C) including comparisons of precursors, intermediary compounds, meteorological conditions and other gaseous/particulate species for two different sites, as presented in our paper, have not been performed before for the western Mediterranean area. More limited comparisons for OA and for one site were done by Chrit et al. (2017); however the comparisons for the Mallorca station are not used in any other articles, therefore are clearly new. Also, articles comparing different SOA simulation schemes have not yet focused on this western part of the basin with its particular set-up and processing different emission sources including continental, coastal, and marine anthropogenic and biogenic emissions.

Source apportionment for the western Mediterranean area is also a subject that has not been much explored before. Of course, source apportionment measurements have

been presented for different sites of the Mediterranean basin (Koçak et al., 2007 ; Querol et al., 2009 ; Minguillón et al., 2011 ; Pey et al., 2013) and also for the European area there are studies that discuss the simulated effects of biogenic emissions (Sartelet et al., 2012) on OA, but none of them give the results of the simulated source apportionment of OA for the whole western basin. This part of our work, answers one of the principal questions raised by the ChArMEx campaign, with the goal of exploring the source apportionment of the OA over the western Mediterranean area. We think that it makes an important contribution to the ACP /AMT special ChArMEx section.

3 – "However, no mention of cloud processing in the aerosol modules is mentioned. Why not link the updated VBS with a cloud processing module?"

While adding cloud processing modules to the model is certainly a good and novel idea, it would be a completely different work apart from the questions that this work seeks to answer to.

4 – "However, these types of evaluations were done during development of those models."

Actually, these schemes were not tested and compared to measurements for the Mediterranean area. They were tested in the US (Robinson et al., 2007 ; Lane et al., 2008), different parts of Europe (Petetin et al., 2014 ; Zhang et al., 2013 ; Fountoukis et al., 2014) and in South America (Shrivastava et al., 2011 ; Hodzic and Jimenez, 2011 ; Shrivastava et al., 2013 ; Shrivastava et al., 2015). Only in Chrit et al (2017) a scheme for the simulation of organic aerosols is tested for the western Mediterranean, but the evaluated scheme is different from the four evaluated schemes in our work (mainly VBS derived schemes in our work versus a one-step surrogate scheme in Chrit et al. 2017).

Concluding remarks:

As a conclusion, our work answers some of the key questions raised by the ChArMEx

project such as evaluation of a CTMs and different organic aerosol schemes and source apportionment of OA in the western Mediterranean basin. It uses for this the wealth of experimental data available from the ChArMEx campaign. We think that these results are novel and important for the valorization of the project. In addition, one of the far reaching goals of the ChArMEx project is to evaluate the modifications in atmospheric composition induced by climate change in this region. Future studies dealing with this issue could build on results of our current study, in terms of evaluation of the CHIMERE and the different organic aerosol schemes. In a second reply and revised version, we will make more ample references to Chrit et al. (2017), which is now published. We will better insist on the novel aspects of our paper as argued in this reply and certainly respond to specific and technical points mentioned in both reviews.

Response written by : Arineh Cholakian et Matthias Beekmann

[revised manuscript text omitted]

---

## Author Comment (AC1) · 8 Feb 2018

In this response, bold parts in quotes are direct extracts of referee comments, blue italic parts are changes made in the article and black normal texts are answers/explanations about each comment made by the referees.

The authors thank the referee for his positive general evaluation of the paper, pointing out its originality and interest. Their detailed comments have allowed its further improvement. The paper has been carefully reread and very long sentences split.

**Referee 1 general comments:**

**1. "Page 6, line 2: I don't understand the aim of the sentence "Four volatility bins are added for anthropogenic and biogenic SOA ranging from 1 to 1000μg.m$^{-3}$". You say on page 5, that you have 9 volatility bins between 0.01 and 10$^6$, corresponding to the figure SI2. On the Figure 2, you represent the 4 volatility bins between 1and 1000. Which bins do you use in the model? Please make the manuscript clearer."**

There is a distinction in this work and also other previous articles concerning VBS between POA volatility bins and ASOA/BSOA volatility bins because of their different nature and also different sources. For ASOA/BSOA species four volatility bins ranging between 1 to 1000μg.m$^{-3}$ are employed, while for POA 9 bins ranging between 0.01 and 10$^6$ are used. In the article, this distinction has been shown in figure 2 for BSOA/ASOA species, while same type of figure for POA is shown in the supplementary materials (Figure SI-1). To show this fact better a text was added to the legend of figure 2:

> *Note that this schematic represents BSOA/ASOA where four bins are used, for SVOC/IVOC (where 9 bins are used) a schematic is presented in SI-1.*

**2. "Page 10, line 34: I don't understand what you mean by "for regimes"."**

The sentence was intended to present meteorological regimes, meaning periods with different meteorological conditions. The wording is changed to "synoptical meteorological conditions" in the revised manuscript.

> *While the general comparison between the meteorological fields used as input for CHIMERE simulations and observations is in general satisfying, when daily averages representative for different synoptical meteorological conditions are compared instead of hourly values, the correlation becomes higher and the bias lower. (Page 11, lines 20-24)*

**3. "Page 12, line 18-20: Could you explain a little more the cloud effect affecting aerosols concentrations."**

The following sentences have been added for the cloud scavenging effect seen in the observations:

> *Cloud scavenging processes are taken into account in the model. However, because of the unique geographical characteristics mentioned before for this site, these fog events were not simulated by the meteorological inputs. Since these decreases concern only a small percentage of the observations, they do not have a major effect on the outcome of these comparisons. While this effect is very visible for sulfates, it is less pronounced for other particulate species such as black carbon and OA. (Page 13, lines 8-13)*

**4. "Page 18, line 10-12: The concentrations of allPOA around Corsica does not reach 1.5μg.m$^{-3}$ in figure 12.g. Same for figure 13.e and the 30% of allPOA to the OA. Could you correct the text?"**

Yes, we agree with the first referee. It has been corrected in the revised version of the article.

**Referee 1 figure related comments:**

1. **"Figure 4: You mention a, b and c, but they are not reported on the figure."**

Yes, it has been corrected in the revised version of the article.

2. **"Figure 5.b1: The blue ribbon around the date 20130726 seems weird. As I understood your method, it should be placed around the simulation results (blue line), but on these part of the graph it seems to be around the black line corresponding to the observation. Could you explain why?"**

This error is rectified in the revised version. The maximum of the ribbon being higher than the limit of the plot, the coding program unfortunately removed the whole upper ribbon for this date.

**Referee 1 technical comments:**

1. **Page 5, line 26: Please add "SI1" after "in the supplementary information".**

Added in the revised version.

2. **Page 6, line 33: as in Shrivastava et al. 2013 -> as in Shrivastava et al. (2013)**

Modified in the revised version.

3. **Page 6, line 36: life time -> lifetime**

Modified in the revised version.

4. **Page 10, line 14: Please add "Orographic Representativeness Error" for the clarity of the text.**

Modified in the revised version.

5. **Page 12, line 30: You say the correlation is 0.26, but in the table it is 0.36 for Ersa and 0.47 for Es Pinar. Could you correct, or explain?**

Modified in the revised version. It was just a confusion between orographic representativeness of 26 % and a correlation of 0.36.

6. **Page 14, line 22: Could you precise what US stands for?**

US stands for United States, it has been changed to *the USA* in the revised version.

Referee 2 comments:
**Referee 2 general comments:**

We wish to thank referee 2 for his evaluation of the paper, which has allowed us to improve the paper. This response integrates our earlier short response to referee 2, which replied to the major issues raised. We hope that our reply can assure the referee that our paper presents indeed original and novel results.

1. **"In addition, the authors do not give appropriate credit of the work of Chrit et al. (2017) who simulated this same data, using a different approach. The authors indicate that the concentrations and properties are well simulated – so this calls into question the need for the current paper. What does this new study tell us that the work of Chrit et al. (2017) did not?"**

Although the two papers are based on results from the same measurements campaign, there are fundamental differences between them. Apart from obvious differences like different models with

different inputs (emissions, meteorological fields …), the approach taken in the two articles regarding the simulation of organic aerosols is clearly different. In our article, we use different VBS schemes, taking into account formation of non-volatile SOA and fragmentation processes in one of them. In Chrit et al. (2017), they use a surrogate based approach where SOA are divided into three types, hydrophilic, hydrophobic or both (Couvidat et Sartelet 2015). In addition, ELVOCs are also taken into account. To our point of view, it is important to compare all these different schemes to detailed measurement data and not limit this comparison to one particular scheme.

In several aspects, our paper goes significantly beyond the Chrit et al. (2017) paper, which was published earlier.

(1) In our work, we use measurements from a second site at Mallorca in addition to data from Cap Corsica for the same period, which is not the case for the work of Chrit et al. (2017). The two sites have different characteristics as it is seen in the results of $^{14}$C measurements and PMF (Positive Matrix Factorization) method. The one at Mallorca is more strongly influenced by anthropogenic emissions. Since the goal of this article is to evaluate the performance of different SOA simulation schemes in the western Mediterranean area, it is more representative to take into account campaign data from these two sites. The use of previously untapped data from a second site at Mallorca not only enriches the comparison, but also opens the research and analysis work to different mixtures of anthropogenic and biogenic emissions, allowing to evaluate and discuss further the capabilities of the model.

(2) Our work provides, in addition to that of OA, a detailed comparison of meteorological inputs, SOA precursors (isoprene and monoterpenes) and intermediate compounds (MVK+MACR) for both sites while Chrit et al. (2017) only focuses on OA.

(3) For the Cap Corsica station, because of its unique geographical characteristics not well represented by CTMs, an orographical representativeness error study is performed in our work. This is actually cited in Chrit et al. (2017) to indicate an error for organic aerosol (Chrit et al. 2017, Page 7, section 4, sub-section 2). The novelty of this approach is discussed in more detail below.

(4) A source apportionment study for the western Mediterranean basin at two different altitudes (model layers between 0-50m and 300-450m above ground) is also included in our study which is not the case in Chrit et al. (2017). Our shows the impact of biogenic and anthropogenic sources for the formation of OA not only on the surface, but also for a higher altitude.

As a side note, the two articles were written at almost the same time period. While Chrit et al. (2017) was accepted for publication on 23 October 2017, this article was submitted for discussion on 26 august 2017. The Chrit et al. (2017) paper being now published, we cited it at several places:

> *For example, Chrit et al. (2017) modelled SOA formation in the western Mediterranean area during the ChArMEx summer campaigns, with a surrogate scheme that also contains ELVOCs (Extremely Low Volatile Organic Compounds). (Page 2, lines 24-26)*

> *Chrit et al. (2017) used a two-step surrogate scheme for the simulation of the Ersa site measurements. They found that their modified SOA simulation scheme corresponds well with the data, with a correlation of 0.67 and a Mean Fractional Bias (MFB) of -0.15 for daily values of the period between June to August 2013. For the period of July to August 2013 for daily values, we find 0.52 correlation and an MFB of -0.03 for the modified VBS scheme, which shows that both these schemes work reasonably well for the simulated area. (Page 19, line 36 - page 20, line 2)*

**2. "As a result of these facts, I do not find this manuscript very novel."**

Here, we wish to show that our work is indeed novel and original in several important aspects. While we appreciate the comments made by referee 2, we strongly disagree with this statement for the reasons specified in detail below.

The Cap Corsica site has unique geographical characteristics which raises issues when comparing simulations to measurements. Indeed, it is located on the northerly edge of a crest and surrounded from the west, north and east by slopes falling rapidly to the sea. Its altitude is at 530 m, while in CHIMERE simulations, even at a 1 km horizontal resolution, it is represented at 360 m height. This height difference might induce differences between simulated and observed concentrations, which need to be assessed. This is even truer in a marine environment with low boundary layer heights. Therefore, a novel approach was developed in this work to calculate an orographic representativeness error. This error expresses the uncertainty in the simulated concentration due to the fact that the orography is not perfectly represented in the model. It is calculated based on the comparison between simulated concentrations at different altitudes (100-500 m) over a narrow domain, with the measurement site at the center. This method allows attributing representativeness errors to different species, which turn out to be large for primary compounds especially emitted in the marine boundary layer like $NO_x$ and BC, and smaller for (mainly) secondary compounds as $O_3$ and OA. To the best of our knowledge, this approach has not been used anywhere else for the estimation of errors produced by orographic representativeness of a site (given that earlier work mostly focusses on problems of horizontal representativeness). This method can be applied to other sites with the same characteristics (high altitude remote sites with strong change of altitude over small distances). The results obtained from this approach are also cited in Chrit et al 2017 (Page 7, section 4, sub-section 2). We would argue that this development, while not the primary goal of the paper, is clearly novel.

For the eastern Mediterranean area, relatively ample literature is available both for aerosol chemical composition (Bardouki et al., 2003 ; Koçak et al., 2007b ; Koulouri et al., 2008 ; Hildebrandt et al., 2010) and also for OA comparisons to simulations (e.g Fountoukis et al., 2014). However, for the western part of the basin, less studies exist for the occidental basin with exception of those done for coastal cities in the area (for example Querol et al., 2009 ; Pey et al., 2013). Furthermore these data have not been used to the best of our knowledge, specifically for CTM model evaluation. Thus, detailed model-observation comparisons with different aspects of OA (total concentrations, oxidation states, modern/fossil fractions from [14]C) including comparisons of precursors, intermediary compounds, meteorological conditions and other gaseous/particulate species for two different sites, as presented in our paper, have not been performed before for the western Mediterranean area. Comparisons of OA for the Cap Corse area was done by Chrit et al. (2017) with a different scheme and a different model; however, the comparisons for the Mallorca station are not used in any other articles, and are therefore clearly new. Also, articles comparing different SOA simulation schemes have not yet focused on this part of the basin, this basin introduces interesting challenges in this area with its particular characteristics and different emission sources including continental, coastal, and marine anthropogenic and biogenic emissions.

Source apportionment studies with simulations for the western Mediterranean area is also a subject that has not been much explored before. Of course, observational based source apportionment analyses have been presented for different sites of the Mediterranean basin (Koçak et al., 2007 ; Querol et al., 2009 ; Minguillón et al., 2011 ; Pey et al., 2013) and also for the European area there are studies that discuss the simulated effects of biogenic emissions (Sartelet et al., 2012) on OA. Our study is original and novel, because it uses a model tied to observations of source apportionment; allowing us to extrapolate results from two sites to a larger area. This part of our work, answers one of the principal goals of the ChArMEx campaign directly, that is exploring the source apportionment of OA over the western Mediterranean area. We think that this part of our paper makes an important contribution to the ACP /AMT special ChArMEx section.

3. **"However, no mention of cloud processing in the aerosol modules is mentioned. Why not link the updated VBS with a cloud processing module?"**

Cloud scavenging processes are already taken into account in the model. However, because of the unique geographical characteristics mentioned before for this site, these fog events were not simulated by the meteorological inputs. Since these decreases concern only a small percentage of the observations, they do not have a major effect on the outcome of these comparisons. While this effect is very visible for sulfates, it is less pronounced for other particulate species such as black carbon and OA.

4. **"However, these types of evaluations were done during development of those models."**

Actually, these schemes were not tested and compared to measurements for the Mediterranean area. They were tested in the USA (Robinson et al., 2007 ; Lane et al., 2008), different parts of Europe (Petetin et al., 2014 ; Zhang et al., 2013 ; Fountoukis et al., 2014) and in South America (Shrivastava et al., 2011 ; Hodzic and Jimenez, 2011 ; Shrivastava et al., 2013 ; Shrivastava et al., 2015). Only in Chrit et al (2017) a scheme for the simulation of organic aerosols is tested for the western Mediterranean, but the evaluated scheme is different from the four schemes evaluated in our work (mainly VBS derived schemes in our work versus a one-step surrogate scheme in Chrit et al. 2017).

**Referee 2 specific comments:**

1. **"Page 4, Line 13. I did not find Table 1 particularly useful. It could be incorporated into the Figure 1 caption. "**

Table 1 has been removed in the revised version of the article, and incorporated in figure 1.

2. **"Page 4, Line 33. What is a shipping snap sector?"**

Shipping SNAP (Selective Nomenclature for Air Pollution) sector is the sector in anthropogenic emissions representing shipping emissions. Some anthropogenic emission inventories use SNAP sectors while others use NFR09 sectors. Both are explained in the references given for MACC-III and Edgar-HTAP emission inventories.  In the revised paper, "snap" is changed to SNAP (Selective Nomenclature for Air Pollution).

3. **"Page 5, Line 10. Has MELCHIOR been updated since its inception in 2003?"**

As a response to this comment, the following text is added to the article:

> *The reaction rates used in MELCHIOR are constantly updated (last update in 2015), however, the reaction scheme itself has not been updated since 2003. Some reactions have been added to it by Bessagnet et al. (2009) regarding the oxidation of organic aerosol precursors, but no official updates have been done. Also, comparison of MELCHIOR has been done with SAPRC-07A, a more recent scheme (Carter, 2010). The results show acceptable differences between MELCHIOR and SAPRC-07A, for example, when compared to EEA ozone measurements, both produce a correlation coefficient of 0.71. These comparisons are presented in Menut et al., (2013), Mailler et al., (2016). (Page 5, lines 12-18)*

4. **"Page 5, Line 13. What is the distribution of particle sizes in the model? That is, the authors provide the range but not any information about how the bins are spaced."**

The following short explanation was added for this comment:

> *This module distributes aerosols in a number of size bins, here 10 bins ranging from 40nm to 40µm, in a logarithmic sectional distribution, each bin spanning over a size range of a factor of two (40-20 µm, 20-10 µm, …). (Page 5, lines 21-22)*

5. **Page 5, Line 24. Pun and Seigneur were not the source of the experimental data. This citation should be to the original manuscripts.**

References are added for experimental data.

*The scheme for simulation of SOA in CHIMERE (Bessagnet et al., 2008) consists of a single-step oxidation process, where VOC lumped species are directly transformed into SVOC (Semi-volatile Organic Compounds) with yields that are taken from experimental data (Odum et al., 1997 ; Griffin et al., 1999 ; Pun and Seigneur, 2007). (Page 5, line 33)*

6. **Page 6, Line 27. Please provide more detail on how this ratio of NO reaction rates is used to determine the low versus high NOx yields. What values of this ratio correspond to low versus high NOx? What are the bounds of this ratio?**

The following explanations are added to the text: *(Page 6, line 34 – page 7, line 5)*

*For this purpose, a parameter ($\alpha$) is added to the scheme, which calculates the ratio of the reaction rate of $RO_2$ radicals with NO ($\nu_{NO}$;high-$NO_x$ regime) with respect to the sum of reaction rates of the reactions with $HO_2$ ($\nu_{HO_2}$), and $RO_2$ ($\nu_{RO_2}$ ;low-$NO_x$ regime).The parameter $\alpha$ is expressed as follows:*

$$\alpha = \frac{\nu_{NO}}{\nu_{NO}+\nu_{HO_2}+\nu_{RO_2}} \qquad\qquad\qquad Eq.\ 1$$

*This $\alpha$ value represents the part of $RO_2$ radicals reacting with NO (which leads to apply "high $NO_x$ yields"). It is calculated for each grid cell by using the instantaneous NO, $HO_2$ and $RO_2$ concentrations in the model. Then, the following equation is used to calculate an adjusted SOA yield using this $\alpha$ value (Carlton et al., 2009).*

$$Y = \alpha \times Y_{highNO_x} + (\alpha - 1) \times Y_{lowNO_x} \qquad\qquad\qquad Eq.\ 2$$

7. **Section 3 Page 7, Line 19. Was only total NOx measured and compared? Were there any issues due to conversion of other NOy species into NO?**

For Ersa, due to its photolytic converter, the CRANOX II instrument measures NO and $NO_2$ without interferences. However, for Es Pinar interferences from $NO_y$ on NO are possible. Figure 5 was modified to include $NO_y$ time series as well as $NO_x$ to address this issue for Es Pinar. The following text is added to the article as a response:

*At Ersa, $NO_x$ (nitrogen oxides) were measured by CraNOx analyzer using ozone chemiluminescence with a resolution of 5 minutes. The photolytic converter in the analyzer allows the conversion of direct measurements of $NO_2$ into NO in a selective way, thus avoiding interferences with other $NO_y$ species. At Es Pinar, an API Teledyne T200 with molybdenum converter was used; therefore, the measurements are not specific to NO and interferences from $NO_y$ species are possible for these measurements. (Page 7, line36 – page 8, line 2)*

Also, table 1 has been updated for the NOx measurement instrument in Es Pinar. In figure 5, $NO_y$ time series are added to the comparisons of simulated and observed $NO_x$ to take into account the possible interference of $NO_y$ in the measurements. Statistics for the comparison of simulated $NO_y$ to $NO_x$ measurements are added to table 3.

8. **Page 8, Lines 2 and 5. I do not understand why a constant value of HOA was assumed for each PMF. This needs to be clarified as model results are later compared to these values.**

A constant value of HOA was not considered for none of the station. The a-value mentioned in the article refers to the extent to which the output HOA factor is allowed to vary from the input HOA reference

mass spectra (i.e. 10% for Ersa and 5% for Cap Es Pinar). The a value is a scalar ranging from 0 to 1 (i.e. 0 to 100%). If the a value is set to 1 then we are in the case of a pure unconstrained PMF approach and if the a value is set to 0 then we are in the case of a Chemical Mass Balance approach (CMB). The use of the a-value approach still accounts for variability of the profile. This method is fully described in Canonaco et al., (2013) and Sturtz, (2014). In such remote environments such as the two stations we are using, the HOA factor could not be extracted from the OA mass spectral matrix with a classical unconstrained PMF. In order to assess the impact of primary fossil fuel combustions, the HOA factor must be constrained using a reference HOA mass spectra. Note that only the HOA factor was constrained, all the other factors being fully free.

*For the Ersa site, HOA (Hydrocarbon-like Organic Aerosol) profile was constrained with a reference HOA factor using an a value of 0.1. The a value refers to the extent to which the output HOA factor is allowed to vary from the input HOA reference mass spectra (i.e. 10% in this case, Canonaco et al., 2013). In such remote environment, the HOA factor could not be extracted from the OA mass spectral matrix with a classic unconstrained PMF approach. Two other factors were extracted, without ant constrains, including SVOOA (Semi-Volatile Oxygenated Organic Aerosol) and LVOOA (Low-Volatile Oxygenated Organic Aerosol). For the Es Pinar site, HOA has been constrained using an a-value of 0.05. Three additional factors were retrieved, including an SVOOA (Semi-Volatile Oxygenated Organic Aerosol) and 2 LVOOA (Low Volatile Oxygenated Organic Aerosol) factors. (Page 8, lines 20-28)*

9. **Section 4 Page 9, Line 6. The wording here should be changed to indicate that it is the simulated height of the cell in which the site was located was underpredicted – not that the simulated height of the site was underpredicted (since the authors are not actually simulating the height of the site).**

Yes, this is true, the phrasing is changed in the article to the following:

*For the 10-km domain (D10), we noticed that there was an inconsistency between simulated and real altitude of the cell where the Ersa site is located. (Page 9, lines 22-23)*

10. **Page 9, Line 27. Higher than a fixed value? What value? What was its basis?**

As mentioned in the article, more detailed explanation about the method of the calculation of ORE (Orographic Representativeness Error) is presented in SI3. The threshold value for correlation between fitted and simulated values mentioned in the article is 0.50, but the method was tested with 0.60 and 0.70 correlations as well, but no significant changes were seen in the resulting ORE for any of the species. The threshold value was chosen somewhat arbitrarily, for assuring satisfying correlation.

11. **Page 9, Line 39. The authors state that secondary species appear less influenced by the orographic uncertainty. However, the impact on MVK + MACR is as large as that for primary species. The authors need to be careful with their word choice.**

This is true, the phrasing of the paragraph should have been more precise. It is changed to the following:

*A general conclusion is that secondary pollutants with higher atmospheric lifetimes appear to be well represented from a geographic point of view. On the contrary, model-observation comparisons for more reactive primary and secondary pollutants with short lifetimes (primary such as $NO_x$ and reactive secondary such as MVK+MACR) should be performed with caution keeping in mind the fact that the simulated altitude is not representative of the orography for this specific station. (Page 10, lines 19-25)*

12. **Page 10, Line 35. I do not believe that this last sentence is necessary.**

Sentence removed.

**13. Page 11, Line 7. The caption from Table 4 needs to refer to Figure 5.**

Figure number corrected.

**14. Page 11, Line 32. Does this requirement of a regional look at the aerosol call into question the use of two sites?**

No, it does not. Since the regional look mentioned here refers to precursors of SOA, meaning isoprene and mono-terpenes (as also mentioned in the article). These two are unstable components with short lifetimes, thus, it is necessary to have a regional look to be able to compare them to simulations more accurately. However, OA has a much longer lifetime, making it much more stable, therefore, two stations can be enough for an accurate observation-simulation comparison.

**15. Page 12, Line 19. Here is evidence for cloud processing (again, it is presented for BC later). However, no mention of cloud processing in the aerosol modules is mentioned. Why not link the updated VBS with a cloud processing module? That would certainly enhance the novelty of the work.**

The following has been added to the text:

*Cloud scavenging processes are already taken into account in the model. However, because of the unique geographical characteristics mentioned before for this site, the meteorological inputs did not simulate these fog events. Since these decreases concern only a small percentage of the observations, they do not have a major effect on the outcome of these comparisons. While this effect is very visible for sulfates, it is less pronounced for other particulate species such as black carbon and OA. (Page 13, lines 8-13)*

**16. Page 13, Line 3. I do not find this summary paragraph necessary.**

The summary paragraph has been removed.

**17. Page 13, Line 27. Figure 7 caption has an error in the word standard.**

The word "standard" has been modified in the figure 7 caption.

**18. Page 13, Lines 30 and 31. Please provide standard deviations on the averages.**

Standard deviation values were added for table 5.

**19. Page 14, Line 12. What is meant by 'respected'?**

The word "respected" is changed to "simulated".

*It is also noticeable that the large-scale tendencies are in most cases simulated in each scheme, therefore different regimes are well predicted by the model. (Page 15, lines 2-3)*

**20. Page 14, Line 34. I do not see much use for Table 7. This text can be included in the main body of the manuscript.**

Table 7 has been removed and its contents incorporated in the manuscript.

*ASOA is considered to be in the fossil fraction and BSOA in the non-fossil fraction. For carbonaceous aerosol, residential/domestic uses are considered as non-fossil as they are mostly related to wood burning (Sasser et al., 2012). Therefore, they are attributed to the non-fossil bin (3.6% for BC and 12.3% for OC, Sasser et al., 2012). The non-fossil contribution of ASOA and POA due to biofuel usage is ignored here, as it is minor (<5%).(Page 15, lines 25-29)*

**21. Page 15 and beyond, The colors in Figures 8, 9, and 11 need to be differentiated to a greater extent if they are included in any future submission.**

White spaces and numbers were added make these images clearer for the reader. But the color pallette was not changed, since the plot separates schemes by their order, and not by their color. The only colors that are of importance in these figures are green in its different shades for non-fossil OA and brown in its different shades for fossil OA.

**22. Page 17, Line 15. I find this summary paragraph unnecessary.**

The summary paragraph is removed.

[revised manuscript text omitted]

---

## Author Response (AR2)

In this response, bold parts in quotes are direct extracts of referee comments, blue italic parts are changes made in the article and black normal texts are answers/explanations about each comment made by the referees and the reviewer. The authors thank both referees and the reviewer for their pertinent comments.

Referee 1 comments:

The authors thank the referee for his positive general evaluation of the paper, pointing out its originality and interest. Their detailed comments have allowed its further improvement. The paper has been carefully reread and very long sentences split.

**Referee 1 general comments:**

**1. "Page 6, line 2: I don't understand the aim of the sentence "Four volatility bins are added for anthropogenic and biogenic SOA ranging from 1 to 1000µg.m$^{-3}$". You say on page 5, that you have 9 volatility bins between 0.01 and 10$^6$, corresponding to the figure SI2. On the Figure 2, you represent the 4 volatility bins between 1 and 1000. Which bins do you use in the model? Please make the manuscript clearer."**

There is a distinction in this work and also other previous articles concerning VBS between POA volatility bins and ASOA/BSOA volatility bins because of their different nature and also different sources. For ASOA/BSOA species four volatility bins ranging between 1 to 1000µg.m$^{-3}$ are employed, while for POA 9 bins ranging between 0.01 and 10$^6$ are used. In the article, this distinction has been shown in figure 2 for BSOA/ASOA species, while same type of figure for POA is shown in the supplementary materials (Figure SI-1). To show this fact better a text was added to the legend of figure 2:

*Note that this schematic represents BSOA/ASOA where four bins are used, for SVOC/IVOC (where 9 bins are used) a schematic is presented in SI-1.*

**2. "Page 10, line 34: I don't understand what you mean by "for regimes"."**

The sentence was intended to present meteorological regimes, meaning periods with different meteorological conditions. The wording is changed to "synoptical meteorological conditions" in the revised manuscript.

*While the general comparison between the hourly meteorological fields used as input for CHIMERE simulations and observations is in general already satisfying, the correlation becomes higher and the bias lower when daily averages representative for different meteorological conditions are compared instead of hourly values. (Page 11, lines 24-26)*

**3. "Page 12, line 18-20: Could you explain a little more the cloud effect affecting aerosols concentrations."**

The following sentences have been added for the cloud scavenging effect seen in the observations:

*Cloud scavenging processes are already taken into account in the model. However, because of the unique geographical characteristics mentioned before for this site, the meteorological inputs did not simulate these fog events and therefore cloud scavenging was not activated in the simulation. Since these decreases concern only a small percentage of the observations, they do not have a major effect on the outcome of these comparisons. While this effect is very visible for sulfates, it is less pronounced for other particulate species such as black carbon and OA. (Page 13, lines 10-15)*

**4. "Page 18, line 10-12: The concentrations of allPOA around Corsica does not reach 1.5µg.m$^{-3}$ in figure 12.g. Same for figure 13.e and the 30% of allPOA to the OA. Could you correct the text?"**

Yes, we agree with the first referee. It has been corrected in the revised version of the article.

**Referee 1 figure related comments:**

1. **"Figure 4: You mention a, b and c, but they are not reported on the figure."**
Yes, it has been corrected in the revised version of the article.

2. **"Figure 5.b1: The blue ribbon around the date 20130726 seems weird. As I understood your method, it should be placed around the simulation results (blue line), but on these part of the graph it seems to be around the black line corresponding to the observation. Could you explain why?"**
This error is rectified in the revised version. The maximum of the ribbon being higher than the limit of the plot, the coding program unfortunately removed the whole upper ribbon for this date.

**Referee 1 technical comments:**

1. **Page 5, line 26: Please add "SI1" after "in the supplementary information".**
Added in the revised version.

2. **Page 6, line 33: as in Shrivastava et al. 2013 -> as in Shrivastava et al. (2013)**
Modified in the revised version.

3. **Page 6, line 36: life time -> lifetime**
Modified in the revised version.

4. **Page 10, line 14: Please add "Orographic Representativeness Error" for the clarity of the text.**
Modified in the revised version.

5. **Page 12, line 30: You say the correlation is 0.26, but in the table it is 0.36 for Ersa and 0.47 for Es Pinar. Could you correct, or explain?**
Modified in the revised version. It was just a confusion between orographic representativeness of 26 % and a correlation of 0.36.

6. **Page 14, line 22: Could you precise what US stands for?**
US stands for United States, it has been changed to *the USA* in the revised version.

Referee 2 comments:
**Referee 2 general comments:**

We wish to thank referee 2 for his evaluation of the paper, which has allowed us to improve the paper. This response integrates our earlier short response to referee 2, which replied to the major issues raised. We hope that our reply can assure the referee that our paper presents indeed original and novel results.

1. **"In addition, the authors do not give appropriate credit of the work of Chrit et al. (2017) who simulated this same data, using a different approach. The authors indicate that the concentrations and properties are well simulated – so this calls into question the need for the current paper. What does this new study tell us that the work of Chrit et al. (2017) did not?"**
Although the two papers are based on results from the same measurements campaign, there are fundamental differences between them. Apart from obvious differences like different models with different inputs (emissions, meteorological fields …), the approach taken in the two articles regarding the simulation of organic aerosols is clearly different. In our article, we use different VBS schemes, taking into account formation of non-volatile SOA and fragmentation processes in one of them. In Chrit et al. (2017), they use a surrogate based approach where SOA are divided into three types, hydrophilic,

hydrophobic or both (Couvidat et Sartelet 2015). In addition, ELVOCs are also taken into account. To our point of view, it is important to compare all these different schemes to detailed measurement data and not limit this comparison to one particular scheme.

In several aspects, our paper goes significantly beyond the Chrit et al. (2017) paper, which was published earlier.

(1) In our work, we use measurements from a second site at Mallorca in addition to data from Cap Corsica for the same period, which is not the case for the work of Chrit et al. (2017). The two sites have different characteristics as it is seen in the results of $^{14}$C measurements and PMF (Positive Matrix Factorization) method. The one at Mallorca is more strongly influenced by anthropogenic emissions. Since the goal of this article is to evaluate the performance of different SOA simulation schemes in the western Mediterranean area, it is more representative to take into account campaign data from these two sites. The use of previously untapped data from a second site at Mallorca not only enriches the comparison, but also opens the research and analysis work to different mixtures of anthropogenic and biogenic emissions, allowing to evaluate and discuss further the capabilities of the model.

(2) Our work provides, in addition to that of OA, a detailed comparison of meteorological inputs, SOA precursors (isoprene and monoterpenes) and intermediate compounds (MVK+MACR) for both sites while Chrit et al. (2017) only focuses on OA.

(3) For the Cap Corsica station, because of its unique geographical characteristics not well represented by CTMs, an orographical representativeness error study is performed in our work. This is actually cited in Chrit et al. (2017) to indicate an error for organic aerosol (Chrit et al. 2017, Page 7, section 4, sub-section 2). The novelty of this approach is discussed in more detail below.

(4) A source apportionment study for the western Mediterranean basin at two different altitudes (model layers between 0-50m and 300-450m above ground) is also included in our study which is not the case in Chrit et al. (2017). Our shows the impact of biogenic and anthropogenic sources for the formation of OA not only on the surface, but also for a higher altitude.

As a side note, the two articles were written at almost the same time period. While Chrit et al. (2017) was accepted for publication on 23 October 2017, this article was submitted for discussion on 26 august 2017. The Chrit et al. (2017) paper being now published, we cited it at several places:

> *For example, Chrit et al. (2017) modelled SOA formation in the western Mediterranean area during the ChArMEx summer campaigns, with a surrogate scheme that also contains ELVOCs (Extremely Low Volatile Organic Compounds). The simulated concentrations and properties (oxidation and affinity with water) of organic aerosols agree well to the observations performed at Ersa (Corsica), after they had included the formation of extremely low volatility organic aerosols and organic nitrate from monoterpenes oxidation in the model. (Page 3, lines 26-28)*

> *Chrit et al. (2017) used a two-step surrogate scheme for the simulation of the Ersa site measurements. They found that their modified SOA simulation scheme corresponds well with the data, with a correlation of 0.67 and a Mean Fractional Bias (MFB) of -0.15 for hourly values of the period between June to August 2013. For the period of July to August 2013 for hourly values, we find 0.52 correlation and an MFB of -0.03 for the modified VBS scheme, which shows that both these schemes work reasonably well for the simulated area. (Page 19, line 20-25)*

**2. "As a result of these facts, I do not find this manuscript very novel."**

Here, we wish to show that our work is indeed novel and original in several important aspects. While we appreciate the comments made by referee 2, we strongly disagree with this statement for the reasons specified in detail below.

The Cap Corsica site has unique geographical characteristics which raises issues when comparing simulations to measurements. Indeed, it is located on the northerly edge of a crest and surrounded from the west, north and east by slopes falling rapidly to the sea. Its altitude is at 530 m, while in CHIMERE simulations, even at a 1 km horizontal resolution, it is represented at 360 m height. This height difference might induce differences between simulated and observed concentrations, which need to be assessed. This is even truer in a marine environment with low boundary layer heights. Therefore, a novel approach was developed in this work to calculate an orographic representativeness error. This error expresses the uncertainty in the simulated concentration due to the fact that the orography is not perfectly represented in the model. It is calculated based on the comparison between simulated concentrations at different altitudes (100-500 m) over a narrow domain, with the measurement site at the center. This method allows attributing representativeness errors to different species, which turn out to be large for primary compounds especially emitted in the marine boundary layer like $NO_x$ and BC, and smaller for (mainly) secondary compounds as $O_3$ and OA. To the best of our knowledge, this approach has not been used anywhere else for the estimation of errors produced by orographic representativeness of a site (given that earlier work mostly focusses on problems of horizontal representativeness). This method can be applied to other sites with the same characteristics (high altitude remote sites with strong change of altitude over small distances). The results obtained from this approach are also cited in Chrit et al 2017 (Page 7, section 4, sub-section 2). We would argue that this development, while not the primary goal of the paper, is clearly novel.

For the eastern Mediterranean area, relatively ample literature is available both for aerosol chemical composition (Bardouki et al., 2003 ; Koçak et al., 2007b ; Koulouri et al., 2008 ; Hildebrandt et al., 2010) and also for OA comparisons to simulations (e.g Fountoukis et al., 2014). However, for the western part of the basin, less studies exist for the occidental basin with exception of those done for coastal cities in the area (for example Querol et al., 2009 ; Pey et al., 2013). Furthermore these data have not been used to the best of our knowledge, specifically for CTM model evaluation. Thus, detailed model-observation comparisons with different aspects of OA (total concentrations, oxidation states, modern/fossil fractions from $^{14}$C) including comparisons of precursors, intermediary compounds, meteorological conditions and other gaseous/particulate species for two different sites, as presented in our paper, have not been performed before for the western Mediterranean area. Comparisons of OA for the Cap Corse area was done by Chrit et al. (2017) with a different scheme and a different model; however, the comparisons for the Mallorca station are not used in any other articles, and are therefore clearly new. Also, articles comparing different SOA simulation schemes have not yet focused on this part of the basin, this basin introduces interesting challenges in this area with its particular characteristics and different emission sources including continental, coastal, and marine anthropogenic and biogenic emissions.

Source apportionment studies with simulations for the western Mediterranean area is also a subject that has not been much explored before. Of course, observational based source apportionment analyses have been presented for different sites of the Mediterranean basin (Koçak et al., 2007 ; Querol et al., 2009 ; Minguillón et al., 2011 ; Pey et al., 2013) and also for the European area there are studies that discuss the simulated effects of biogenic emissions (Sartelet et al., 2012) on OA. Our study is original and novel, because it uses a model tied to observations of source apportionment; allowing us to extrapolate results from two sites to a larger area. This part of our work, answers one of the principal goals of the ChArMEx campaign directly, that is exploring the source apportionment of OA over the western Mediterranean area. We think that this part of our paper makes an important contribution to the ACP /AMT special ChArMEx section.

**3. "However, no mention of cloud processing in the aerosol modules is mentioned. Why not link the updated VBS with a cloud processing module?"**

Cloud scavenging processes are already taken into account in the model. However, because of the unique geographical characteristics mentioned before for this site, these fog events were not simulated by the

meteorological inputs. Since these decreases concern only a small percentage of the observations, they do not have a major effect on the outcome of these comparisons. While this effect is very visible for sulfates, it is less pronounced for other particulate species such as black carbon and OA.

4. **"However, these types of evaluations were done during development of those models."**

Actually, these schemes were not tested and compared to measurements for the Mediterranean area. They were tested in the USA (Robinson et al., 2007 ; Lane et al., 2008), different parts of Europe (Petetin et al., 2014 ; Zhang et al., 2013 ; Fountoukis et al., 2014) and in South America (Shrivastava et al., 2011 ; Hodzic and Jimenez, 2011 ; Shrivastava et al., 2013 ; Shrivastava et al., 2015). Only in Chrit et al (2017) a scheme for the simulation of organic aerosols is tested for the western Mediterranean, but the evaluated scheme is different from the four schemes evaluated in our work (mainly VBS derived schemes in our work versus a one-step surrogate scheme in Chrit et al. 2017).

**Referee 2 specific comments:**

1. **"Page 4, Line 13. I did not find Table 1 particularly useful. It could be incorporated into the Figure 1 caption. "**

Table 1 has been removed in the revised version of the article, and incorporated in figure 1.

2. **"Page 4, Line 33. What is a shipping snap sector?"**

Shipping SNAP (Selective Nomenclature for Air Pollution) sector is the sector in anthropogenic emissions representing shipping emissions. Some anthropogenic emission inventories use SNAP sectors while others use NFR09 sectors. Both are explained in the references given for MACC-III and Edgar-HTAP emission inventories. In the revised paper, "snap" is changed to SNAP (Selective Nomenclature for Air Pollution).

3. **"Page 5, Line 10. Has MELCHIOR been updated since its inception in 2003?"**

As a response to this comment, the following text is added to the article:

*This mechanism has around 120 reactions to describe the whole gas-phase chemistry. The reaction rates used in MELCHIOR are constantly updated (last update in 2015), however, the reaction scheme itself has not been updated since 2003. Some reactions have been added to it by Bessagnet et al. (2009) regarding the oxidation of organic aerosol precursors, but they don't affect gas-phase chemistry. Also, MELCHIOR has been compared to SAPRC-07A, a more recent scheme (Carter, 2010), and the results show acceptable differences between the two schemes, for example, when compared to EEA (European Economic Area) ozone measurements, both produce a correlation coefficient of 0.71. These comparisons are presented in Menut et al. (2013) and Mailler et al. (2017). (Page 5, lines 13-19)*

4. **"Page 5, Line 13. What is the distribution of particle sizes in the model? That is, the authors provide the range but not any information about how the bins are spaced."**

The following short explanation was added for this comment:

*This module distributes aerosols in a number of size bins, here 10 bins ranging from 40nm to 40µm, in a logarithmic sectional distribution, each bin spanning over a size range of a factor of two (40-20 µm, 20-10 µm, …). (Page 5, lines 21-23)*

5. **Page 5, Line 24. Pun and Seigneur were not the source of the experimental data. This citation should be to the original manuscripts.**

References are added for experimental data.

*The SOA simulation scheme in CHIMERE (Bessagnet et al., 2008) consists of a single-step oxidation process, where VOC lumped species are directly transformed into SVOC (Semi-volatile Organic Compounds) with yields that are taken from experimental data (Odum et al., 1997 ; Griffin et al., 1999 ; Pun and Seigneur, 2007). (Page 5, line 33-35)*

**6. Page 6, Line 27. Please provide more detail on how this ratio of NO reaction rates is used to determine the low versus high NOx yields. What values of this ratio correspond to low versus high NOx? What are the bounds of this ratio?**

The following explanations are added to the text: *(Page 7, lines 1-8)*

*For this purpose, a parameter (α) is added to the scheme, which calculates the ratio of the reaction rate of $RO_2$ radicals with NO ($v_{NO}$;high-$NO_x$ regime) with respect to the sum of reaction rates of the reactions with $HO_2$ ($v_{HO_2}$), and $RO_2$ ($v_{RO_2}$ ;low-$NO_x$ regime).The parameter α is expressed as follows:*

$$\alpha = \frac{v_{NO}}{v_{NO}+v_{HO_2}+v_{RO_2}} \qquad\qquad Eq.\ 1$$

*This α value represents the part of $RO_2$ radicals reacting with NO (which leads to apply "high $NO_x$ yields"). It is calculated for each grid cell by using the instantaneous NO, $HO_2$ and $RO_2$ concentrations in the model. Then, the following equation is used to calculate an adjusted SOA yield using this α value (Carlton et al., 2009).*

$$Y = \alpha \times Y_{highNO_x} + (\alpha - 1) \times Y_{lowNO_x} \qquad\qquad Eq.\ 2$$

**7. Section 3 Page 7, Line 19. Was only total NOx measured and compared? Were there any issues due to conversion of other NOy species into NO?**

For Ersa, due to its photolytic converter, the CRANOX II instrument measures NO and $NO_2$ without interferences. However, for Es Pinar interferences from $NO_y$ on NO are possible. Figure 5 was modified to include $NO_y$ time series as well as $NO_x$ to address this issue for Es Pinar. The following text is added to the article as a response:

*At Ersa, $NO_x$ (nitrogen oxides) were measured by CraNOx analyzer using ozone chemiluminescence with a resolution of 5 minutes. The photolytic converter in the analyzer allows the conversion of direct measurements of $NO_2$ into NO in a selective way, thus avoiding interferences with other $NO_y$ species. At Es Pinar, an API Teledyne T200 with molybdenum converter was used; therefore, the measurements are not specific to $NO_2$ and interferences of $NO_y$ are possible for these measurements. (Page 8, lines 1-5)*

Also, table 1 has been updated for the NOx measurement instrument in Es Pinar. In figure 5, $NO_y$ time series are added to the comparisons of simulated and observed $NO_x$ to take into account the possible interference of $NO_y$ in the measurements. Statistics for the comparison of simulated $NO_y$ to $NO_x$ measurements are added to table 3.

**8. Page 8, Lines 2 and 5. I do not understand why a constant value of HOA was assumed for each PMF. This needs to be clarified as model results are later compared to these values.**

A constant value of HOA was not considered for none of the station. The a-value mentioned in the article refers to the extent to which the output HOA factor is allowed to vary from the input HOA reference mass spectra (i.e. 10% for Ersa and 5% for Cap Es Pinar). The a value is a scalar ranging from 0 to 1 (i.e. 0 to 100%). If the a value is set to 1 then we are in the case of a pure unconstrained PMF approach and if the a value is set to 0 then we are in the case of a Chemical Mass Balance approach (CMB). The use of the a-value approach still accounts for variability of the profile. This method is fully described in Canonaco et al., (2013) and Sturtz, (2014). In such remote environments such as the two stations we are using, the HOA factor could not be extracted from the OA mass spectral matrix with a

classical unconstrained PMF. In order to assess the impact of primary fossil fuel combustions, the HOA factor must be constrained using a reference HOA mass spectra. Note that only the HOA factor was constrained, all the other factors being fully free.

*For the Ersa site, HOA (Hydrocarbon-like Organic Aerosol) profile was constrained with a reference HOA factor using an a value of 0.1. The a value refers to the extent to which the output HOA factor is allowed to vary from the input HOA reference mass spectra (i.e. 10% in this case, Canonaco et al., 2013). In such remote environment, the HOA factor could not be extracted from the OA mass spectral matrix with a classic unconstrained PMF approach. Two other factors were extracted, without ant constrains, including SVOOA (Semi-Volatile Oxygenated Organic Aerosol) and LVOOA (Low-Volatile Oxygenated Organic Aerosol). For the Es Pinar site, HOA has been constrained using an a-value of 0.05. Three additional factors were retrieved, including an SVOOA (Semi-Volatile Oxygenated Organic Aerosol) and 2 LVOOA (Low Volatile Oxygenated Organic Aerosol) factors. (Page 8, lines 23-31)*

9. **Section 4 Page 9, Line 6. The wording here should be changed to indicate that it is the simulated height of the cell in which the site was located was underpredicted – not that the simulated height of the site was underpredicted (since the authors are not actually simulating the height of the site).**

Yes, this is true, the phrasing is changed in the article to the following:

*For the 10-km domain (D10), we noticed that there was an inconsistency between simulated and real altitude of the cell where the Ersa site is located. (Page 9, lines 29-31)*

10. **Page 9, Line 27. Higher than a fixed value? What value? What was its basis?**

As mentioned in the article, more detailed explanation about the method of the calculation of ORE (Orographic Representativeness Error) is presented in SI3. The threshold value for correlation between fitted and simulated values mentioned in the article is 0.50, but the method was tested with 0.60 and 0.70 correlations as well, but no significant changes were seen in the resulting ORE for any of the species. The threshold value was chosen somewhat arbitrarily, for assuring satisfying correlation.

11. **Page 9, Line 39. The authors state that secondary species appear less influenced by the orographic uncertainty. However, the impact on MVK + MACR is as large as that for primary species. The authors need to be careful with their word choice.**

This is true, the phrasing of the paragraph should have been more precise. It is changed to the following:

*A general conclusion is that secondary pollutants with higher atmospheric lifetimes appear to be well represented from a geographic point of view. On the contrary, model-observation comparisons for more reactive primary and secondary pollutants with short lifetimes (primary such as NOx and reactive secondary such as MVK+MACR) should be performed with caution keeping in mind the fact that the simulated altitude is not representative of the orography for this specific station. (Page 10, lines 27-31)*

12. **Page 10, Line 35. I do not believe that this last sentence is necessary.**

Sentence removed.

13. **Page 11, Line 7. The caption from Table 4 needs to refer to Figure 5.**

Figure number corrected.

14. **Page 11, Line 32. Does this requirement of a regional look at the aerosol call into question the use of two sites?**

No, it does not. Since the regional look mentioned here refers to precursors of SOA, meaning isoprene and mono-terpenes (as also mentioned in the article). These two are unstable components with short lifetimes, thus, it is necessary to have a regional look to be able to compare them to simulations more

accurately. However, OA has a much longer lifetime, making it much more stable, therefore, two stations can be enough for an accurate observation-simulation comparison.

**15. Page 12, Line 19. Here is evidence for cloud processing (again, it is presented for BC later). However, no mention of cloud processing in the aerosol modules is mentioned. Why not link the updated VBS with a cloud processing module? That would certainly enhance the novelty of the work.**

The following has been added to the text:

*Cloud scavenging processes are already taken into account in the model. However, because of the unique geographical characteristics mentioned before for this site, the meteorological inputs did not simulate these fog events and therefore cloud scavenging was not activated in the simulation. Since these decreases concern only a small percentage of the observations, they do not have a major effect on the outcome of these comparisons. While this effect is very visible for sulfates, it is less pronounced for other particulate species such as black carbon and OA. (Page 13, lines 10-15)*

**16. Page 13, Line 3. I do not find this summary paragraph necessary.**

The summary paragraph has been removed.

**17. Page 13, Line 27. Figure 7 caption has an error in the word standard.**

The word "standard" has been modified in the figure 7 caption.

**18. Page 13, Lines 30 and 31. Please provide standard deviations on the averages.**

Standard deviation values were added for table 5.

**19. Page 14, Line 12. What is meant by 'respected'?**

The phrase was removed.

**20. Page 14, Line 34. I do not see much use for Table 7. This text can be included in the main body of the manuscript.**

Table 7 has been removed and its contents incorporated in the manuscript.

*ASOA is considered to be in the fossil fraction and BSOA in the non-fossil fraction. For carbonaceous aerosol, residential/domestic uses are considered as non-fossil as they are mostly related to wood burning (Sasser et al., 2012). Therefore, they are attributed to the non-fossil bin (3.6% for BC and 12.3% for OC, Sasser et al., 2012). The non-fossil contribution of ASOA and POA due to biofuel usage is ignored here, as it is minor (<5%).(Page 15, lines 12-16)*

**21. Page 15 and beyond, The colors in Figures 8, 9, and 11 need to be differentiated to a greater extent if they are included in any future submission.**

White spaces and numbers were added make these images clearer for the reader. But the color pallette was not changed, since the plot separates schemes by their order, and not by their color. The only colors that are of importance in these figures are green in its different shades for non-fossil OA and brown in its different shades for fossil OA.

**22. Page 17, Line 15. I find this summary paragraph unnecessary.**

The summary paragraph is removed.

Reviewer comments:
Thank you

**1. p3 , line 38 at at**

Fixed in the final version

2. **p5, line 16 don't -> do not**

Fixed in the final version

3. **Please avoid acronyms inside figure captions, give the definition.**

Image legends were revised and explanations are added for all acronyms.

4. **I recommend to do something very simple such as adding one paragraph to introduce the focus on western Mediterranean, the use of the Orographic Representativeness Error and the consideration of factors other than OA for model evaluation.**

The following phrases were added to the introduction:

*This enables us to perform model-observation comparisons with unprecedented detail over this region, including not only the OA concentration, but also its origin (14C analyses), its oxidation state (PMF method results). In addition, a comparison for meteorological parameters and gaseous/particulate species which could affect the production of OA or could help analyze the robustness of the used schemes has been performed. Moreover, because of the orographic complexity of one of the sites (Ersa, Cap Corse) explored in this work, a novel method is designed to calculate the orographic representativeness error of different species. To the best of our knowledge, this is the first time that the concentrations of precursors, intermediary products and OA concentrations and properties have been simulated for different OA simulation schemes and compared for each scheme to multiple series of measurements in different stations for the western Mediterranean area.*

Also, a phrase was added to the abstract to explain about ORE.

*Because of the complex orography of the Ersa site, an original method for calculating an orographic representativeness error (ORE) has been developed.*

[revised manuscript text omitted]

**Figure 12.** Average oOrganic aerosol (OA) concentrations over the western Mediterranean basin simulated from 10 July to 9 August 2013. Simulations were performed with the modified VBS scheme. Left column is near the surface, and right column for an altitude ~between 300-450 m asl. From top to bottom: a-b: total OA concentration (µg.m⁻³); c-d: BSOA (Biogenic secondary OA) concentration (µg.m⁻³); e-f: total ASOA (Anthropogenic secondary OA concentration) concentration (µg.m⁻³); g-h: sum of POA (Primary OA) and its subsequent oxidations products (µg.m⁻³). Results are from the D3 simulation (3-km horizontal resolution) within the framed area, from the D10 simulation outside (10-km resolution).

[Figure]

**Figure 13.** As figure 12 but for relative contributions of BSOA/ASOA/POA to total OA (%). From top to bottom: a-b: biogenic secondary organic aerosol (BSOA); c-d: anthropogenic secondary organic aerosol (ASOA); e-f: POA and its subsequent oxidation products.